# "It Feels Like My Spine is About to Break": Experience and support needs of family caregivers of children with cerebral palsy in Ethiopia

Melkitu Melak[1]*, Solomon Mekonnen[2], Afolasade Fakolade[1], Beata Batorowicz[1]

**1** School of Rehabilitation Therapy, Queen's University, Kingston, Ontario, Canada, **2** Institute of Public Health, College of Medicine and Health Sciences, University of Gondar, Gondar, Ethiopia

* 20mfm9@queensu.ca

## Abstract

### Background

Due to the complex and long-term care needs of children with Cerebral Palsy (CP), caregivers often face an overwhelming caregiving burden, and they experience physical and psychological strain. Few studies in Sub-Saharan Africa have indicated that caregivers face financial difficulties and limited access to formal support services. However, evidence is scarce in Ethiopia.

### Objectives

This study aimed to explore the caregiving experiences and support needs of family caregivers of children with CP in Ethiopia.

### Method

An exploratory descriptive qualitative design was used in this study. A purposeful sampling method was employed to identify and select 13 family caregivers of children with CP. Face-to-face, semi-structured, in-depth individual interview was used to collect data. The interview was conducted in the local language, Amharic. The audio data were transcribed verbatim manually and imported to NVivo version 14 software for analysis. Reflexive thematic analysis was used to analyze the data.

### Results

Four themes were identified: the emotional journey of caregivers, daily caregiving demands, impact of caregiving, and support systems and support needs. Caregiving for children with CP negatively affected the emotional, physical and social well-being of caregivers. Caregivers indicated a need for financial support, psychological

**Data availability statement:** The data consists of transcripts of the qualitative interviews with family caregivers of children with cerebral palsy and is majority in Amharic. Because there are only 13 participants, we are not permitted to include to make de-identified data available in order to protect the confidentiality of our participants. The manuscripts contained direct citations from the participants and is the data that we publicly share.

**Funding:** This work was supported by a doctoral scholarship from the Mastercard Foundation Scholars Program at the University of Gondar (to MM) covering only data collection expenses but not publication fees. The funder had no role in the study design, data collection and analysis, decision to publish, or manuscript preparation.

**Competing interests:** The authors have declared that no competing interests exist.

support, adequate healthcare service, access to assistive devices, and education for their children with CP.

## Conclusion

The findings emphasize the urgent need for interventions that address the financial vulnerabilities of caregivers, provide targeted psychosocial support, and enhance access to healthcare, assistive devices and education for children with CP.

---

## Introduction

Cerebral palsy (CP) is one of the most common developmental disabilities, having been recognized as a distinct clinical condition since the mid-19th century [1]. According to Rosenbaum et al. (2007), CP is characterized by permanent movement and posture disorders resulting from non-progressive disturbances in the developing brain, leading to activity limitations and participation restrictions [2].

The global incidence of CP is approximately 1 in 500 births, with an estimated 18 million individuals affected worldwide [3]. In High-Income Countries (HICs), the prevalence rate of CP is about 1.6 per 1,000 live births, while in Low and Middle-Income Countries (LMICs), the rate is between 3 and 3.4 per 1,000 live births [4,5]. A study of the Global LMICs CP register indicated that 79.2% of children with CP had spastic CP, with 73.3% classified at Gross Motor Function Classification System (GMFCS) levels III-V, which indicates moderate to severe CP [6]. In Ethiopia, hospital-based data from Addis Ababa indicate that 60.4% of children with CP have spastic quadriplegia, often accompanied by speech difficulties (95.4%), cognitive impairments (87.4%), epilepsy (60.9%), and sensory impairments [7]. However, national-level data on CP prevalence in Ethiopia remain unavailable due to the absence of a CP registry or surveillance system, although it is known that 17.6% of the country's population lives with a disability [8].

CP often necessitate long-term care beyond the typical needs of children without disabilities, placing a considerable burden on family caregivers, particularly mothers, who typically provide day-to-day support [9]. Evidence showed that the most significant predictors of caregiver burden were the duration of caregiving and the level of care recipient dependency [10,11]. Caregivers of children with CP in Turkey were found to spend an average of 15 hours per day on caregiving activities [12], and caregivers in Ethiopia spend 21.3 hours per day on caregiving activities [13]. As the child with CP gets older and heavier, the physical demands on caregivers also increase [14]. Furthermore, mothers of children with CP with higher scores on the GMFCS tend to spend more time on caregiving than mothers of children with lower GMFCS scores [15]. These caregivers also experience higher levels of stress due to increased caregiving demands, health concerns, and potential developmental delays.

Evidence mainly from HICs indicated the burden of caregiving faced by mothers of children with CP, who often experience physical pain, emotional and psychological strain [16–19]. These adverse outcomes of caregiving are exacerbated in the

absence of support [20]. Support enables caregivers to provide necessary care at home and helps them function effectively [21], and in HICs, caregivers often have access to formal support systems that are tailored to their specific needs and goals [22,23]. Although family caregiving can be challenging, it also offers rewards such as a deep sense of fulfillment and satisfaction. In line with this, caregivers of children with CP from Australia, United States of America (USA) and United Kingdom (UK) reported feelings of personal growth, strengthened social networks, a heightened sense of purpose, and a deepened sense of meaning in life, which enhance their overall quality of life [24,25].

Studies in LMICs also showed that mothers of children with CP frequently experience physical health issues such as headaches and backaches, as well as psychological conditions like stress, anxiety, and depression [26,27]. Economic hardship, limited access to formal support services such as respite care, and inadequate infrastructure for accessible transportation are common concerns highlighted in research from Sub-Saharan Africa (SSA) [14,28–31]. Moreover, societal attitudes such as social stigma and discrimination were also reported to contribute to the psychosocial challenges caregivers face [32]. Disability is often misunderstood or associated with superstition in many African communities, leading to social isolation and reduced community support [33]. Caregivers frequently reported feelings of guilt, helplessness, and overwhelming concern about their children's future due to the lack of societal understanding of CP [30,34]. This isolation, combined with the physical and emotional demands of caregiving, can contribute to significant mental health burdens, such as anxiety, depression, chronic stress, and even suicidal ideation [35], especially when there is a lack of support [36,37].

A recent scoping review on the health outcomes and support needs of caregivers of children with CP in SSA highlighted a lack of research from Ethiopia, despite a growing body of evidence in the region [38]. While previous studies from other countries could provide some insight, the unique socio-cultural and economic factors, such as the spiritual and collectivist nature of Ethiopian society [39], multidimensional poverty [40], and community members negative attitude towards children with disability [41] might influence the caregiving experience differently. Moreover, in Ethiopia, there is limited availability and accessibility of disability support systems, including community-based and specialized rehabilitation services, and healthcare services [42,43], educational services for children with disability [33,44], lack of disability or caregiver benefits [33], and day care and respite care services [45,46]. These challenges were linked by researchers to increased physical, financial and emotional demands of caregiving and were found to negatively impact the overall wellbeing of caregivers [13,47]. To address the existing knowledge gap related to Ethiopia's context, the current study explored the caregiving experiences and support needs of family caregivers for children with CP in Ethiopia. Such a study is needed to provide a foundational evidence base to inform future research, as well as give insight into the design of relevant policies and support interventions tailored to the specific needs and realities of caregivers of children with CP in Ethiopia.

## Materials and methods

### Study design

This work is part of a PhD research project of the first author under the supervision of the last author. In this study, an exploratory descriptive qualitative approach was used, which is recommended when little is known about the phenomenon [48]. The experience and support needs of family caregivers of children with CP have received little attention in the Ethiopian context; as a result, we adopted an exploratory descriptive qualitative approach to gain authentic insight into the experiences of family caregivers.

### Study setting

To better understand the experiences and support needs of caregivers of children with CP, it is vital to understand the social, cultural, and health system context of Ethiopia. Ethiopia is the second most populated country in Africa and the 10th largest in the world, with a population of 132,938,555 in 2025 [49]. According to the 2007 census, 83.6% of Ethiopia's

population resides in rural areas. Ethiopia has the fastest-growing economy, yet it remains one of the poorest countries, with a per capita gross national income of $1,020 [49].

Ethiopia is an uncolonized country and has a rich and unique cultural heritage. In the Ethiopian context, culture and religion promote altruism and support for others as a fulfillment of religious or spiritual obligations, which in turn influence societal practices [39]. Women and girls often assume caregiving responsibilities within family settings. Mothers are traditionally expected to be the primary caregivers and household caretakers [50], and there is a high cultural expectation for mothers to care for children with disabilities at home. However, the stigma surrounding disability is widespread in Ethiopia, where it is often viewed as a divine punishment or a curse from ancestors or the result of evil spirits [51,52]. Spiritual healing practices like "Holy Water" are vital for many Ethiopians who follow the Christian faith, especially those suffering from chronic health issues.

The study was conducted in Gondar, located in the Amhara Region, northwest Ethiopia, approximately 745 km from the capital, Addis Ababa. Gondar is situated in the foothills of the north mountain chains at an average elevation of 2,300 meters above sea level, characterized by its mountainous topography. It was a royal capital city, which is now rich in Orthodox Christian heritage, and a cultural hub in the Amhara region. At the commencement of this study, there was civil war in the Amhara region, including Gondar and its surrounding areas. The instability of the situation and related safety issues made it difficult to recruit participants from rural areas.

Since 2010, Ethiopia's health service delivery has been structured into three tiers, providing primary, secondary, and tertiary levels of healthcare [53]. The primary level of healthcare unit includes health posts, health centers and primary hospitals. Health posts health service delivery is at the very bottom level of an administrative unit called a "kebele," serving approximately 5,000 people. Five satellite health posts are attached to a referral health center, providing services to approximately 25,000 people, which deliver promotive, preventive and basic curative services. A primary hospital provides emergency surgical services, as well as inpatient and ambulatory services, to an average population of 100,000, and serves as a referral center for health centers. At the secondary level of the healthcare system, a general hospital serves as a referral centre for primary hospitals and serves about 1–1.5 million people at the district level. At the tertiary level, a specialized hospital is a referral centre for general hospitals and serves 3.5–5 million people at the regional and central levels. Most healthcare services are funded through out-of-pocket payments [53,54].

World Health Organization (WHO) recommends integrating medical rehabilitation services across all levels of the healthcare system, particularly at the primary healthcare level [55]. However, in Ethiopia, the integrated rehabilitation services are scarce and primarily limited to the tertiary level of the healthcare system, mainly providing physiotherapy services [56]. Affordability and transportation were found to be common barriers to accessing medical rehabilitation services among people with disabilities in Ethiopia [40]. Community-based rehabilitation (CBR) was first launched by WHO as a strategy to increase access to rehabilitation services at the community level for persons with disability [55]. The University of Gondar (UoG) CBR program is the only community-based rehabilitation program serving Gondar City and its 14 surrounding districts. It was established in 2005 in partnership with Light for the World and Save the Children International. It provides home-based physical rehabilitation, assistive devices and referral to advanced medical care services for persons with disabilities (https://uog.edu.et/cbr/).

## Study participants and recruitment

Study participants were family caregivers of children with CP in Gondar. We employed purposeful sampling to identify and select participants, considering the age of the child and the severity of CP to capture and describe the phenomenon in various situations [57]. Family caregivers were included if the following criteria were fulfilled: (a) primary family caregiver aged ≥18 who provided most of the care in a day for the child with CP aged <18 years, (b) speaks the local language, Amharic, (c) providing care for a child with CP with GMFCS levels IV or V (indicating severe forms of CP), and (d) willing to participate in the study. First, we recruited 15 caregivers of children with CP from the UoG CBR logbook. However, two

of those caregivers did not meet the criteria because their children did not have severe CP (i.e., GMFCS level IV and V), and they did not take part in the study. Caregivers who were recruited for the study were contacted by CBR field workers to inquire about their willingness to participate in the research.

Consistent with an exploratory, descriptive, qualitative approach, sample size adequacy was guided by the concept of information power which indicates the more information the sample holds relevant for the study, the fewer participants is needed [58,59]. The size of a sample with sufficient information power depends on five factors: (a) the aim of the study, (b) sample specificity, (c) use of established theory, (d) quality of dialogue, and (e) analysis strategy [59]. For the current study, the aim of this study was to gain an understanding of the caregiving experience of family caregivers of children with CP in Ethiopia, sample included only primary family caregivers of children with severe CP, the Multidimensional Model of Caregiving Process and Outcome guided the study [20], and the method of analysis was a thematic cross-case approach [59]. To enhance the quality of dialogue during data collection, participants were selected purposively, considering their ability to articulate independent ideas in Amharic language. Therefore, the aim of the study was narrow, samples were specific, conceptual framework was used, and quality dialogue was achieved between the interviewer and the caregivers which all indicated that a small sample size is needed for our study. We decided to stop at 13 participants, as the data we obtained were rich and sufficient to answer our research question.

## Data collection

Data collection took place at a newly opened daycare center for children with disabilities, a familiar place for family caregivers, from mid-March to May 2024. First, prior to the commencement of the interview, the interviewer completed a background questionnaire to gather information on sociodemographic characteristics, including age, sex, relationship with the child, ethnicity, residence, educational status, employment status, marital status, and child characteristics. GMFCS, MACS, and CFCS tools were used to classify the functional ability of children with CP as perceived by caregivers while guiding them on the meaning of each level. Then, face-to-face, semi-structured, in-depth individual interviews were conducted in Amharic, the local language. Each interview was audio recorded, with an average length of 40.5 minutes, ranging between 28 and 53 minutes. The interview guide was developed based on the Multidimensional Model of Caregiving Process and Outcome, developed by Raina et al. (2004), which provides a comprehensive understanding of the complex interplay between various factors that influence the caregiving experience. This model highlights how background/contextual factors (e.g., socioeconomic status), caregiver strain, intrapsychic factors and coping factors interact to affect caregiver outcomes such as physical and psychological health. The model informed the development of the interview guide, helping us identify key areas to explore, including caregivers' day-to-day experiences, challenges, coping strategies, available support systems, and the broader contextual influences on caregiving [20]. The interviewer was a local female sociologist with extensive experience in qualitative research. A three-hour training session was conducted by the first author (MM) for the data collector, covering the study's aims, interview guide, procedures, and ethical guidelines. The data collector began the conversation with each participant by providing a brief overview of the study's purpose. Next, warm-up questions were asked, followed by the main questions, as outlined in the interview guide (S1 File). Participants were provided an honorarium to cover their transportation and lunch costs. During and immediately after each interview, the interviewer wrote field notes to capture non-verbal cues and her reflections. A debriefing meeting with MM was conducted via a phone call.

## Data analysis

The Amharic audio data were transcribed verbatim manually and imported to NVivo software. Reflexive thematic analysis was used to analyze the data [60,61]. This analysis approach aligns with the study's theoretical and paradigmatic foundations, which underlines the subjectivity of participants' experiences and acknowledge the reflexive influence of the researcher's interpretation in shaping the analysis [62]. We followed six phases of thematic analysis, which reflect

recursive and iterative processes [63]. First, to gain a better contextual understanding of the data, MM familiarized herself with the data by repeatedly listening to the audio-recorded data, reading the Amharic transcripts, and reviewing the field notes. Then, MM completed open-coding in the Amharic transcripts using NVivo 14 (QSR International) software [64]. AB (secondary coder) coded one of the transcripts and discussed with MM to develop the initial codes. In addition, one transcript was translated into English, and MM and BB (senior author) discussed the initial codes, as well as held multiple discussions during the development of categories, sub-themes, and themes. Themes were generated by aggregating codes with similar meanings, translating them into English with the associated codes and data, and sharing them with the research team for feedback. Then, a recursive review of candidate themes was conducted based on the coded data items and the entire dataset. Finally, themes and sub-themes were defined in relation to the dataset and the research question, and then the report was produced [61,65]. To maintain consistency in meaning, themes, subthemes, and associated data were translated from Amharic to English by a professional translator and then back to Amharic by two language-proficient translators with a health science background. Descriptive analysis was completed for sociodemographic data.

## Trustworthiness

As Lincoln and Guba (1985) recommended, we ensured rigor by establishing four criteria: credibility, dependability, conformability, and transferability. To ensure credibility, the first author repeatedly listened to the audio data and read the Amharic transcripts several times to prolong the engagement with the data. A secondary coder coded some of the data to get multiple perspectives (triangulation). In addition, BB and MM continuously discussed the research process and findings, which were critically evaluated by the entire research team [66].

An audit trail was maintained to enhance dependability by documenting all aspects of the research process. NVivo supported the audit trail of the coding process by systematically documenting each coding activity, making the process transparent and accessible. Reflexive journaling was used to record reflections on methodological decisions and challenges, as well as how these influenced the study's progress. To ensure confirmability, the first author engaged in self-reflection to acknowledge how background, values, experiences, and assumptions could impact the data analysis and interpretation, allowing the presentation of the findings in a balanced and honest manner. A detailed description of the research context and participants is provided to strengthen the transferability of the findings.

## Ethical considerations

We obtained ethical clearance from Queen's University Health Sciences & Affiliated Teaching Hospitals Ethics Board (Ref.No-6039390) and also from the University of Gondar, Institutional Ethics Review Board of the College of Medicine Health Sciences, and Specialized Hospital (Ref. No-CMHSHRCS-06/02/05/9/2023). The research process, purpose, potential risks, and benefits were explained to all participants in their local language, Amharic, and all participation was voluntary. Written informed consent was obtained from the participants, and the data collector assisted by reading the consent form aloud to those who lacked sufficient literacy to read it independently. To maintain confidentiality, individual identifiers were removed, and participants were given number identifiers.

## Inclusivity in global research

Additional information regarding the ethical, cultural, and scientific considerations specific to inclusivity in global research is included in the Supporting Information (S1 Checklist).

## Sociodemographic characteristics of family caregivers

A total of 13 family caregivers of children with CP were interviewed, of whom 92.3% (n = 12) were mothers. The mean age of family caregivers was 34.5 years, ranging between 27 and 45 years. All family caregivers were Orthodox Christians. More than half of family caregivers were unemployed (Table 1). About 23% (n = 3) of family caregivers were divorced

**Table 1. Sociodemographic characteristics of family caregivers of children with CP.**

| Age in years | n=13(%) |
|---|---|
| 25-29 | 4(30.8) |
| 30-34 | 2(15.4) |
| 35-39 | 5(38.3) |
| 40-45 | 2(15.4) |
| **Relationship with the child** | |
| Mother | 12(92.3) |
| Aunt | 1(7.7) |
| **Residence** | |
| Urban | 12(92.3) |
| Rural | 1(7.7) |
| **Marital status** | |
| Married | 9(69.2) |
| Divorced | 3(23.1) |
| Widowed | 1(7.7) |
| **Religion** | |
| Orthodox Christian | 13 (100) |
| **Level of education** | |
| No education | 2(15.4) |
| Primary | 7(53.8) |
| Secondary | 2(15.4) |
| College | 2(15.4) |
| **Economic status as perceived by caregivers (poor, medium, rich)** | |
| Poor | 12(92.3) |
| Medium | 1(7.7) |
| **Number of children in the household** | |
| ≤2 | 9(69.2) |
| 3-4 | 4(30.8) |
| **Occupation** | |
| No | 6(46.1) |
| Cleaner | 4(30.8) |
| Daily labourer | 2(15.4) |
| Farmer | 1(7.7) |

following the birth of their child with CP. Regarding the children's characteristics, 69.2% (n=9) of children with CP were female. The mean age of children with CP was 7.1 years, ranging between 2.83 and 16 years. Of the 13 children with CP, only one was enrolled in school. (Table 2).

## Findings

### The experience and support needs of family caregivers of children with CP

Four themes were identified from the analysis of transcripts: emotional journey of caregivers, daily caregiving demands, impact of caregiving, and support systems and support needs. These themes, along with their subthemes and representative participant quotes, are presented in Table 3 and described below.

**Table 2. Characteristics of children with CP.**

| Age in years. Months | n = 13 (%) |
|---|---|
| 2-4 | 3(23.1) |
| 4.1-6 | 4(30.8) |
| 6.1-12 | 5(38.3) |
| 12.1-18 | 1(7.7) |
| **Biological sex** | |
| Male | 4(30.8) |
| Female | 9(69.2) |
| **Birth order relative to siblings** | |
| 1st | 3(23.1) |
| 2nd | 6(46.1) |
| 3rd | 2(15.4) |
| 4th | 2(15.4) |
| **Education enrollment** | |
| Enrolled | 1(7.7) |
| Not enrolled | 12(92.3) |
| **GMFCS*** | |
| IV | 6(46.1) |
| V | 7(53.8) |
| **MACS**** | |
| I | 3(23.1) |
| II | 1(7.7) |
| IV | 2(15.4) |
| V | 7(53.8) |
| **CFCS*** ** | |
| II | 1(7.7) |
| III | 11(84.6) |
| V | 1(7.7) |

*Gross Motor Function Classification System

**Manual Activity Classification System

***Communication Function Classification System

## Theme 1. Emotional journey of caregivers

Caregivers reported experiencing various emotional reactions from the time of the initial diagnosis across all other challenges brought by caregiving for their child with CP. This theme was defined by sub-themes of feeling shocked, regretting and worrying about the child's future.

**Feeling shocked.** Caregivers shared how they started to show concern when their child didn't reach the developmental milestones at the expected age. At the time of the diagnosis, caregivers experienced shock, disbelief, and an overwhelming sense of grief and confusion. One participant said: "I was shocked when I first heard of my child's condition (P-9)". Another participant explained:

> When she couldn't sit and walk like her friends, I was concerned and took her to the hospital; they told me it was CP caused by complications during childbirth. Hearing this explanation, I was fainted because she was normal when she was born. (P-3)

**Table 3. Themes and sub-themes.**

| Number | Themes | Sub-themes |
|---|---|---|
| 1 | Caregivers emotional journey | Feeling shocked |
| | | Feeling regret |
| | | Worrying about the child's future |
| 2 | Daily caregiving demands | Providing practical daily support |
| | | "We are their tongue"- Supporting child's communication |
| | | "I couldn't make it work"-Balancing caregiving with life dynamics |
| | | Transportation and environmental obstacles |
| | | Social stigma |
| 4 | Impact of caregiving | Impact on physical health |
| | | Impact on mental health |
| | | Social isolation |
| | | Personal growth and resilience |
| 5 | Support systems and support needs | Natural support |
| | | "I used to believe this was only happening to me"-CBR support |
| | | Support needs |

Caregivers expressed feelings of unpreparedness and a lack of knowledge about the actions they could take immediately following the diagnosis. They experienced a sense of hopelessness regarding their child's prognosis and contemplated abandonment as a potential escape; however, none of the participants ultimately considered institutionalizing their children with CP. For example, one mother said:

When doctors told me it was CP, I didn't understand what it was. I thought the child was of no use, and my family suggested giving him up for adoption. I didn't believe that he could improve. I was concerned about where to leave or what to do with him. (P-2)

**Feeling regret.** As caregivers navigated the challenging journey of supporting a child with CP, they often found themselves grappling with deep feelings of regret about the opportunities that slipped away and the dreams that were left unfulfilled. They reflected on the moments that could have been, contemplating how different their lives might have been had their child been born healthy or, perhaps, if they had passed away shortly after birth: "Sometimes, I wish she had passed away at birth; then, I wouldn't have suffered this much." (P-5). These thoughts weighed heavily on their hearts, intertwining with their daily experiences and shaping their perceptions of what could have been a more straightforward, hopeful life. One caregiver described:

I wanted to go to and work in an Arab country and return. If my child were healthy, I would have left him with my family and gone there. The reason I am living in poverty now is because of my child's disability. (P-2)

Other participants said: "I had a different goal in the past and abandoned it, now living low where I am from; I can't escape it. How would you escape?" (P-4). Caregivers also described feeling trapped and wishing they could turn back the clock. One mother indicated:

People around me used to wonder where I would end up, with positive expectations. However, I now find myself feeling stuck in this situation. I prayed to God, asking Him to change the course of time for me. (P-7)

**Worrying about the child's future.**  This sub-theme reflects the emotional burden of caregivers regarding the future of their child with special needs. Caregivers were concerned about how they would carry and lift when the child grows, how to manage menstruation for females, and when they could no longer care. For example, one mother explained her worries as follows:

I am deeply concerned about her future. If she is not cured, it won't be easy. Since she is female, she will experience menstruation after the age of 14, and I am worried about how I am going to manage that. Additionally, I am concerned about how I will carry her as she grows. I ask myself: Will my child be in this situation forever? What will I do? How can I manage her needs? These are my concerns about the future. Right now, she is young, and caregiving is not as challenging as I expect it will be in the future. The future will be much more challenging and overwhelming compared to the present. (P-3)

Another caregiver described her concerns about her daughter becoming an adult and how she wishes to live as long as her daughter to be able to take care of herself:

What if I die in the middle of this? Who will take care of her? Who will make sure she has food and water? No one will care for her like a mother does. I pray for our death to come the same day. In addition, she is 16 years old and going through puberty. I feel uncomfortable when I go out and leave her alone because I'm worried about the possibility of her being assaulted by local gangs. (P-6)

Still, other mothers talked about many other uncertainties related to independence of their children. For example, one mother said:

I'm worried about his future. Will he be able to speak? Will he be able to walk? Will he use his hands? What's causing his difficulty with swallowing? What's obstructing his esophagus? All these uncertainties are weighing heavily on my mind. (P-2)

### Theme 2: Daily caregiving demands

This theme encompasses the daily tasks and responsibilities of caregivers for children with CP, along with the external challenges they encounter. Five subthemes define this theme, which are: providing practical daily assistance, supporting child communication, balancing caregiving with life dynamics, transportation and environmental obstacles, and social stigma.

**Providing practical daily assistance.**  This sub-theme describes the essential practical assistance that family caregivers provide to their children with CP in their daily lives. It outlines various tasks that caregivers undertake, such as preparing meals, feeding, assisting with personal hygiene (toileting, bathing, dressing), engaging in play, carrying (transporting) the children, and managing their medications. Family caregivers reported how they were committed to attending frequent medical checks and physical therapy appointments by carrying their children with CP on their backs for long distances. Furthermore, the caregivers described feeding and dressing as challenging as the child couldn't sit upright. One mother said:

I wake up in the early morning and prepare her breakfast. I wipe her face, toilet her, and feed her breakfast. After three or two hours, there's another meal and her medicine. The challenge is that it takes her two hours to feed in the morning (…) At lunchtime, I give her another meal. I keep an eye on the time, making sure she doesn't miss toileting. Often, she wets herself. So, I must change her clothes before someone else comes and sees her in that state (…) I go to the

hospital or health center every week or two. She spends a lot of time at the hospital—three times a week, I take her carrying on my back. (…). It's a struggle, but I am doing it. There's no transportation, but I come from the outskirts of [city] three times a week carrying her on my back and walking. (P-13)

Another mother explained how the increased muscle tone of the child with CP made the daily caregiving difficult: "I bathe her frequently and feed her on my lap because she can't sit upright. Dressing her is also challenging due to the stiffness in her legs." (P-8)

**"We mothers are their tongue"- Supporting child's communication.** This sub-theme illustrates the role of family caregivers in facilitating communication for children with CP who struggled to express themselves by speech, emphasizing the caregivers' essential role in ensuring the child is heard and understood. The phrase "We are their tongues" signifies how those caregivers spoke on behalf of their children, interpreting their needs, feelings, and desires by understanding children's facial expressions, body movements or other cues. One mother described:

(…) because a child like this can't speak, **we moms are their tongue**, or we are the one who understands what their needs are through their cues, my child never says a word when he is thirsty or hungry, he can't say it, we mothers are the ones who can understand and express their need with much struggle. (P-7)

Participants described how they interpreted children's communication signs, expressions, vocalizations and body movements. One mother noted:

If she is thirsty, she tried to say a word near water, and when she wanted something to eat, she protruded out her tongue; when she wanted to pee, she kept slipping from her buttock. She has her cue, and I understand her cues because I am her mom living with her all the time. (P-3)

Mothers also talked about their hopes and attempts to communicate despite lack of children's ability to speak. One mother stated:

I raise my voice, like an echo from a tin can, trying to get her attention, and I call her loudly, saying, "Mom, love!" even if she doesn't respond. I keep doing it because I believe that over time it will help her learn to pay attention and listen, even though speaking is hard for her. (P-13)

**"I couldn't make it work" - Balancing caregiving and life dynamics.** This subtheme describes the sacrifices caregivers make to manage the demands of caregiving alongside other life commitments such as marriage and work. It emphasizes how caregivers prioritize the comfort and well-being of their children over their own needs. For example, one mother captured the hard situation she faced as she needed to choose between her husband and her child. She said:

(…) I couldn't manage both (marriage responsibility and caregiving). I thought it would be better to prioritize my child, so I divorced my husband. If I had a healthy child, I might have been able to balance everything. But for me, it isn't easy to handle both. I couldn't make it work. (P-1)

Another mother explained the time needed to care for her child and needed sacrifices:

When raising a child like this, you sacrifice much of yourself. You could get married, of course, but it might not work out. If I start a relationship, I won't be able to give my child the time he needs. So, I want to waste my time and energy with my child. I've thought about it and decided to focus on my child and leave my life behind completely. (P-7)

Participants also explained how they needed to quit working to be able to care for their children. For example, one mother said:

> I used to have a job, but because of him, I had to leave it. It was my business; I was a hairdresser and had a salon. Everything stopped. Those who used to work for me (crying….) have now reached a great place in these six years, while it's been challenging for me. (P-1)

**Transportation and environmental obstacles.** Caregivers expressed the challenges of accessing the only available daycare services due to the long distance from home to Gondar and difficulties with transportation. They desired more conveniently located daycare options. One of the mothers explained: "I cannot bring my child to daycare due to the distance I live from Gondar and the difficulty of getting transportation. I wish the daycare were closer so that I could use it" (P-13). In addition, caregivers who had access to a wheelchair preferred to carry their child on their back instead of using the wheelchair for transportation due to the geographically challenging terrain and road conditions that made wheelchair use impossible. One mother said: "Despite that I had a wheelchair provided by the CBR, I prefer to carry my 12-year-old daughter on my back because the road condition is downhill and not suitable for wheelchair use." (P-12)

**Social stigma.** This subtheme reflects the lack of understanding, and negative attitudes of the community members and the caregiver's own family towards children with CP, and their caregivers. The participants shared that others had misconceptions about the cause of CP and understood it as a punishment from God or caused by a curse. Community members feared the potential negative consequences of being in contact with children and their belongings. Caregivers were insulted and blamed as they were made responsible for the child's disability. One mother explained:

> My own families keep asking why the child can't walk. They say, "Is this a curse? Or what have you done for this to happen? 'What have you done to God?" They think it's something I brought up by myself, and they fear a generational curse. They don't seem to have any understanding of what's happening. (P-9)

Another mother shared how she was insulted by others: "People around me perceive that CP is caused due to a curse or punishment from God in response to my wrongdoing; they insult me like this, and it hurts me a lot." (P-3). In addition, some participants indicated that they lost meaningful social interactions due to misconceptions and stigma. For example, one participant said: "My neighbours withdraw me from any social interaction due to my child's disease condition; they no longer share anything with us, even fire, which is the minimum thing we used to share." (P-6)

### Theme 3. Impact of caregiving

This theme encompasses the impact of the demand for caregiving on the health and social life of family caregivers, as well as the benefits of caregiving. The subthemes that defined this theme were the impact on physical health, mental health, social isolation, and personal growth and resilience.

**Impact on physical health.** Participants emphasized how caregiving was physically demanding over time due to the increased age and weight of the child, which created difficulty in lifting and carrying. They described the exhaustion, intense pain, numbness, and lack of sleep as the result of the demanding nature of caregiving. One mother said: "Lifting to transfer and carrying her on my back is challenging as her age and body weight increase. Now, I feel completely exhausted, and my back hurts so badly." (P-10)

Another caregiver explained how she developed back pain due to taking care of her child:

> I am taking care of her, but I'm struggling and feeling exhausted. It's affecting my health, and most of the time, my back hurts. Given my age and physical condition, carrying her on my back from home to the hospital is overwhelming. I can't

sit comfortably or lie down on my side without feeling intense pain in my back. **It feels like my spine is about to break** (…). (P-13)

Caregivers also shared consequences of physical work to lift and carry their child over time, as they became older. One mother explained:

For the past three years, I've been holding him on my lap because nothing works to calm his cries. My left leg, in particular, no longer functions properly. I rely on a taxi for transportation and struggle with walking downstairs. My hands have become numb from holding him for so long. At night, he has difficulty falling asleep. I hold him on my lap and try to comfort him, but he often stays awake until 2:00 a.m., and I can't sleep until he does. (P-4)

Another caregiver said: "Wherever I go, I used to carry her on my back up to her 14 years old, but now I can't manage to carry her; my body is exhausted." (P-6)

**Impact on mental health.** This sub-theme highlights the mental health challenges faced by caregivers of children with CP. Caregivers described experiencing feelings of anger, frustration, anxiety, forgetfulness, and difficulty concentrating, which stemmed from the physical and emotional demands of their role. They also continuously worried about their child's future. Witnessing their child's struggles firsthand, coupled with societal misconceptions and stigma surrounding disabilities, added to the mental health strain experienced by caregivers. For example, one mother talked about her life as 'chaotic' and her effort to seek solitude, while she also needed to care for other children:

(…) caregiving for my son significantly impacted my mental health. I haven't had any rest, and it's taking a toll on me. I've been experiencing forgetfulness lately, and my life feels chaotic and unstable. I seek solitude, often shutting myself in at home to find peace. However, my other kids' constant noise and interruptions make maintaining the silence I need difficult. (P-7)

Other caregivers indicated feeling angry and overwhelmed. One participant explained:

(…) I feel angry about why this is happening to me. Lately, I've started forgetting where I put things and losing track of my thoughts. This happens because I'm overwhelmed by the physical demands of caregiving and constantly worrying about what will happen to her tomorrow. (P-13)

Seeing the pain that the child experiences daily and the stigma surrounding the disability have affected the caregiver's mental health, as captured by this mother:

Even though this disability affects the child, it has a significant impact on the mental health of me. This is because, first, I experience my child's pain every day, and second, the community's misconceptions and insults about disability also cause mental suffering. Some people even believe that CP is a result of the parent's sins, and hearing this can be devastating and cause immense mental anguish. (P-5)

**Social Isolation.** Caregiving impacted the ability of family caregivers to engage in social activities. Some caregivers uncovered that they lacked assistance with childcare, making it challenging to participate equally in social life with their peers. Others shared that they spent considerable time caring for their children with CP, which led to feelings of isolation, even from close family members. Concerns regarding their child's safety, including medication management and personal care, further limited their attendance at social gatherings. Even those with some support for childcare often struggled with a lack of trust in others to provide the same level of care and attention, resulting in a disconnection from social activities.

Participants described losing contact with their peers. One mother explained:

I can't connect with anyone because it takes time to do these things. It's hard to take my son to social activities. I can't leave him at home as no one else cares for him (…). I don't have any social relations. So, I can't participate in anything, and I can't be equal with my peers. (P-1)

Caregivers described how bringing their child to social events made them worry about humiliation, so they stopped attending such events. One mother said: "Participating in social life is challenging for me, and I live in isolation, even from my family. I spend most of my time caring for her. If I take her with me, I worry about facing stigma" (P-6). As participants emphasized their social isolation, they also identified a lack of adequate social support as a source of their isolation:

I am unable to take her to social events such as funerals, weddings, or visits to the sick. I have no one to leave her with, and I worry about potential hazards, such as exposure to electricity or hot water. This leaves me isolated from social life. If I go somewhere, who will give her medication on time? Who will feed her with the same care I do? I wouldn't have any peace of mind if I left her alone. I wouldn't trust anyone else to handle her toileting and personal hygiene. As a result, I remain disconnected from social life. (P-13)

**Personal growth and resilience.** Despite the negative impact of caregiving on health and social life, family caregivers also discovered numerous positive effects of caring for a child with CP. Caregiving opened the door to enhancing social interaction skills, exploring new career opportunities, and experiencing significant growth in maturity and a sense of purpose through the evolving role and relationship with the child. Caregiving has also provided chances to establish meaningful social connections, deepened their faith in God, and helped develop a resilience mindset for future life challenges. One caregiver explained:

Previously, I struggled with interacting with people and didn't know how to handle social situations. However, due to my child and thanks to the training support I received from CBR, I am much better at this. I can now engage with others more effectively and feel more comfortable in social interactions. I also got a job opportunity at CBR. (P-3)

Participants also mentioned how caregiving has helped them build social connections and access valuable emotional support, which has brought them happiness and gratitude. One mother said:

Due to my child, I know a lot of people. I sell fruits on the street, and people often come by to buy from me, not just for the fruit but to support and encourage me. While the money helps, what matters most is the emotional support I receive. It makes me feel happy and grateful. (P-5)

Some caregivers also mentioned how they matured after having a child with CP. One mother said: "After I gave birth to her, she changed my life and made me a mature person" (P-9). Participants expressed that the presence of their child has profoundly shaped their life and identity. One mother said: "I thank God because, without my child coming into my life, I would not know what I would be like" (P-1). In addition, caregivers emphasized that taking the child to different spiritual and religious places questing for healing helped them to strengthen their faith. One mother explained:" I didn't know God before, but because of my child, I've had the opportunity to visit various spiritual and religious places that have strengthened my faith. I am very grateful and happy for this experience." (P-3).

Caregivers explained how they learned valuable lessons from their continued challenges, which have strengthened their ability to be prepared for future adversities and fostered resilience. One mother said:

Even if I don't foresee greater difficulties beyond what I've already faced, I have been taught that it's possible to encounter more hardship. I understand that more challenges might come, and I am prepared to face them with the lesson I got from this experience. (P-4)

One caregiver used an Ethiopian proverb, "Horns are not heavy for a cow," (P-2), which indicated that certain burdens or responsibilities may feel natural or manageable for those who bear them, just like a cow carrying its horns. In the context of caregiving for a child, it reflected her determination and love as a parent, indicating that while carrying her daughter may be physically demanding, it's a part of her life that she embraces without complaint.

## Theme 4. Support systems and support needs

**Natural support.** Caregivers received physical, informational, and emotional support from their natural support systems, such as family, friends, and neighbours. However, the same support sources were reported as a source of distress in the earlier findings. Spirituality and religion played a crucial role in the lives of family caregivers, with all caregivers emphasizing that their faith in God helped them find the strength and support needed to persevere. Engaging in spiritual practices, such as attending church and praying, served as vital coping strategies during difficult times, alleviating worries and enhancing their overall well-being.

Caregivers highlighted how siblings, friends, and neighbours were essential sources of support through their caregiving journeys; however, these circles were reported as a source of distress in the earlier findings. One mother said: "Her older sister helps me a lot, especially when her period is coming; she prepares and changes her underwear and pad when needed." (P-6). However, the involvement of siblings in caregiving negatively impacted their school attendance and educational achievement. For instance, one caregiver noted, "Whenever I want to go somewhere, my daughter has to miss school to care for her sibling with CP. As a result, her academic performance is very low."(P-8).

Another mother indicated how friends were supportive emotionally and led to a formal support service: "My friends are on my side all the time by giving advice and reassurance comfort, and telling me where to go to get services, and they are the ones who told me about CBR and pushed me to enroll." (P-2). Similarly, one mother expressed the emotional support she got from her neighbours: "My neighbours, even though they didn't help me with childcare, provided me encouragement and empathy and gave me hope by saying, 'One day he is going to be fine. Be strong'." (P-4).

All caregivers mentioned that religion and spirituality helped them cope with the stress of caregiving, and they attributed divine intervention as the source of their endurance and survival. One mother explained:

As an Orthodox Christian, I find peace through prayer, which brings me tranquility and helps me feel better. I prefer going to church when I am under significant stress, as it helps alleviate my worries. This practice has always been my way of coping with stress. (P-1)

Another mother expressed that no one was on her side except God who helped her to come this far:

Despite all the challenges, I thank God that I am still surviving. It is through God's help that I have come this far. I now realize that my strength, power, and endurance all come from the child of the Virgin Mary. No one else has been with me by my side; only God stands and strengthens me. (P-7)

Spiritual fathers also provided emotional support for some caregivers by offering home visits, advice, and check-in calls, ensuring they did not miss weekly church masses. One mother said:

My spiritual father gave advice and reassured me that cerebral palsy is not a sign of divine anger. I faithfully attend mass every Sunday; if I miss a service, my spiritual father always checks in on me. He also makes sure my child receives holy water and Holy Communion. Their care and concern for both of us bring me great happiness. (P-3)

While caregivers initially experienced emotional pain and sorrow, they ultimately came to embrace their role with a deeper understanding and acceptance. Caregivers conveyed that their child was a gift from God. They explained how spirituality helped them to overcome the struggle and accept a new reality. One mother said:

Initially, I was overwhelmed. However, as time passed, I had to calm myself, accept the situation, and thank God. That's all I could do... (crying) I realized there was no way out through struggle, so I chose to be grateful to God. I accepted it, and now I pray that He (God) doesn't add more challenges (sobbing). That's how I've managed to get through this, through prayer and trusting in God. (P-1)

Caregivers described how faith helped them to accept their child's condition. One mother explained:

In our community, people point out and say these are mothers of children with disabilities. Now, it doesn't resonate. Because I didn't buy it from the market or choose it, God gave it to me. It would have resonated with me if it had been in the past, and I was crying a lot. (P-9)

Another caregiver indicated: "There is nothing I can do because God gave him to me. After all, the Creator gave him to me" (P-12). Similarly, another mother said: "I am grateful; she is God's gift. I will not go anywhere, no choice; I have accepted what God has given me." (P-5). Caregivers reported being overwhelmed by the caregiving and turned to God to seek a cure for their child and to get relief from the caregiving burden. One mother explained: "What can I do when God has given me this burden? I wish He grant her the cure." (P-13).

**"I used to believe this was only happening to me" - CBR support.** CBR was the only source of formal support for caregivers, and it offered various forms of assistance, including education, small financial loans for some caregivers, physiotherapy, wheelchairs, and daycare services for their children. CBR also facilitated the formation of support groups and regular meetings, fostering a sense of community that alleviated feelings of isolation, and promoted emotional adjustment through shared experiences. Furthermore, those who accessed the daycare reported experiencing mental relief, and their children showed positive progress. One mother said:

"I have never gotten support from the government or any other organization except CBR. CBR supported me a wheelchair for my child and a small loan to start a business, but this doesn't match with my needs." (P-1).

Caregivers who had been attending "Holy Water" for a long time were now thankful for their child's progress following physical therapy given by CBR field workers. One mother indicated:

I believed holy water would heal her, and I tried holy water in different places for an extended period. Then, lately, I took her to the hospital, where I heard about the physical therapy offered by CBR. Even though she couldn't speak so far, I am grateful to see some progress, such as holding a bottle and feeding herself. I am genuinely thankful. (P-13)

The participants highlighted how CBR organized caregivers into support groups and delivered training regularly. Caregivers were able to accept their child and got relief after meeting other caregivers with the same condition at CBR. One mother explained:

Before I met other caregivers like me, **I used to believe this was only happening to me**. But now, after seeing children with even more severe disabilities here in CBR, I've come to accept her with gratitude to God. I am thankful for her, and I embrace her as a blessing. (P-8)

Another caregiver explained how CBR provided training, which improved her mental well-being, however, she could not get financial support, which was her priority. This mother said:

I used to feel deeply distressed when I saw her typically developing peers. Over time, the training and education have helped me feel more at ease and improved my mental well-being. However, beyond that, CBR has not offered any financial support, which I need most. (P-3)

The daycare in CBR helped caregivers to get relief from the caregiving burden and children also benefited, for example, they started gaining weight after joining the daycare. One mother noted:

I used to cry when I saw my son's age equals going to school, but now I got mental relief since my child has joined this daycare; he is also gaining weight because they feed him. I am so grateful for those who made this happen. (P-2)

**Support needs.** All caregivers underscored the importance of caregiver-focused support to ensure the well-being of their children with CP. One mother said:

Although the impairment affects the children, it is the parents who suffer the most and are deeply impacted. Primarily, I believe it is essential to focus on supporting the parents rather than the children. So, the main focus should be on supporting parents. If parents are supported, they can provide better care and protection for their children. When parents are healthy and have a good mental health, they can better care for and love their children. (P-1)

Caregivers reported various support needs, including financial support, psychological counseling, healthcare services, access to assistive devices, and access to child education. The financial challenges faced by caregivers affected their ability to meet the essential daily needs of their children, particularly for items such as milk, diapers, and food, as captured by one caregiver who said: "For my child, all the basic things are needed because I am financially unable to fulfill them by myself." (P-1). Another mother stated: "I couldn't afford to pay rent. I need housing support. If I do not pay rent, then I can sleep with peace of mind." (P-2). Consequently, the main support need identified by caregivers was financial support to foster economic independence and self-sufficiency. One participant explained:

Even though I'm struggling financially a lot, I'm not asking for money from people as an option. I don't want to beg people. I want to work and improve my life while caring for my child, but I lack the financial resources to start a small business. I applied for a government loan, which was denied because I don't have property to use as collateral. Therefore, I am seeking financial support to help me get started. (P-3)

Caregivers couldn't leave their children to work, but they wanted to run a small business from home while caring for their kids. Financial issues were a big obstacle. One participant said: "I would be very happy if I could start working while caring for my child. I believe that working is the only way to solve my financial struggles. I have always thought of having a small kiosk. However, without a penny on my hand, it remains nothing more than a wish." (P-1). Another participant added: "If I get financial support, I will create my own business at home." (P-6). The ongoing conflict and instability in the region have exacerbated their economic challenges, along with the rising cost of living. One caregiver explained: "Due to the war, I lost my source of income. At times, we didn't even have anything to eat. I started a small business on the street

to meet some of my children's needs, but as you can see, the situation [the inflation and instability] in the country has worsened. It's hard to earn any income from that now." (P-7).

In addition, the psychological counselling services were identified as essential in addressing caregivers' mental health needs, thereby enhancing their capacity to provide optimal care for children. One mother explained:

Caregivers like me should get psychological counselling because our mental well-being is greatly affected. I need support for my mental health, and CBR should have professionals who can offer this crucial psychological support. I have not received any psychological support so far. (P-7)

The caregivers highlighted their unmet need to obtain adequate healthcare services. Key issues mentioned by the participants included a lack of comprehensive health information about CP, long waiting times, unprofessional attitudes from professionals, and an absence of essential treatments or therapies. Caregivers expressed their lack of knowledge about the cause of their child's condition, treatment and its long-term implications due to unclear and inadequate information from healthcare professionals. One mother explained:

I don't even understand the illness's nature and cause. Whenever I ask the doctors about his condition, they keep telling me it is related to birth, that is it. I am not educated to navigate about the disease. I want to know if it is a problem related to the brain and how it can be resolved. These were my concerns, but what should I do if the doctors couldn't clarify? How can I know by myself? I couldn't. (P-4)

Due to a lack of knowledge about CP, some caregivers blamed themselves, associating CP with wrongdoing and punishment from God. One mother said, "I questioned God why He gave me a child like this, wondering what sin I had committed" (P-8).

Long waiting times and a poor attitude from health professionals, led caregivers to stop follow-up appointments and place hope in divine intervention for their child's condition. One mother described:

I have been bringing my child to the hospital for his follow-up appointments. Despite arriving early in the morning, I often wait the entire day, with no priority given to him. (…) One day, the doctor stopped me and asked why I was coming. She said, 'He is no longer growing, so there's no need to come anymore.' After that, I stopped the follow-up. I began praying, 'God, you are the one who can restore everything you've given me.' I realized there was no point in continuing to go to the hospital, so I stopped the follow-up. (P-7)

In addition, caregivers reported that physical therapy was the only treatment provided for their children with CP in the hospital, which didn't bring improvement in their children's condition. As a result, they were obliged to seek alternative cure, such as holy water for healing. For example, one mother indicated:

I have taken her to the hospital three or four times, but it hasn't helped because she received no treatment beyond physiotherapy. I will take her to the "holy water" if God grants her forgiveness; I have no other options (…). I remain hopeful that God will bring her healing. (P-11)

Caregivers identified a need for assistive devices such as wheelchairs for transportation and helping their child to sit and feed in an upright position. One mother explained:

My child needs a wheelchair. She wants to sit, but I don't have a wheelchair, can't afford to buy. I made a seat from a cardboard box and clothes, but she's growing and [the seat] is no longer strong enough to hold her. If I could find a wheelchair, it would make a big difference and allow her to sit comfortably. (P-3)

                                                                                18 / 27

Furthermore, caregivers emphasized the importance of education for their children. One mother said: "My child needs an education. I want her to go to school like her friends and learn and transform her life." (P-10). Apart from the long-term benefit that education brings to the child's future, caregivers believed that sending their child to school would provide some time for them to work and meet their child's needs. One mother noted: "If my child can go to school, I will also be able to work and meet her needs." (P-6).

## Discussion

The aim of this study was to explore the caregiving experiences and support needs of family caregivers of children with CP in Ethiopia. Caregiving for children with CP was found to be a complex experience characterized by demanding responsibilities, external challenges and a limited support system that negatively impacted the well-being of caregivers, despite some positive experiences being reported. Financial support, psychological support, adequate healthcare service, access to assistive devices, and access to child education were the support needs identified by caregivers. Below, we discuss the main findings in relation to the existing evidence and provide suggestions for future research, policy, and practice.

### The experience of caregiving for children with CP

Globally, deep-rooted gender norms result in family caregiving still being widely perceived as women's work [67]. As a result, women currently perform up to 81% of unpaid caregiver roles worldwide [68]. In the current study, all caregivers were women (12 were mothers and 1 was an aunt) who assumed the responsibility of meeting their children's daily needs, managing household chores, and sometimes serving as primary breadwinners. Our participant representation reflected the deeply rooted patriarchal norms in Ethiopia, which expect women, specifically mothers, to carry the primary burden of domestic duties, including caring for children with disability [69]. In this context, caregiving could impose a double burden on women who already have demanding day-to-day responsibilities and might also exclude them from participation in education, paid employment, and political and social engagement [46,70]. Such exclusion could reduce their access to resources, autonomy in household and healthcare decision-making, and impact their well-being, ultimately affecting their ability to provide quality care for their child with CP. Contrary to our study, a recent research conducted in Saudi Arabia found that caregiving for children with CP was shared between mothers and fathers, indicating a shift from the traditional view of women as the primary caregivers [71]. Although the study purposefully searched for fathers, our study participant representation may highlight the need to increase awareness to challenge prevailing gender norms in unpaid care.

On top of this, caregivers in our study frequently attended hospital visits and physical therapy sessions for their children's complex health issues. They also faced stigma, limited access to assistive devices, a lack of respite care services, and accessible physical infrastructure. Many of them carried their children on their backs regardless of the children's age and weight, which exacerbated physical health concerns such as exhaustion, back pain, numbness, and sleep disturbances. The findings align with a study conducted in India, which highlighted that caregivers of children with CP experienced a profound burden resulting from the intersection of gender inequity, poverty, and social stigma [72]. Moreover, the conceptual model of the caregiving process and outcome indicated that the caregiver's well-being is a dynamic interaction between contextual factors, the care recipient's functioning, and available support systems. Therefore, more demanding caregiving roles, coupled with inadequate support systems, lead to negative physical and psychological health outcomes for caregivers, which indicates the concepts underpinning in this model are relevant to the Ethiopian context [9,20]. Previous studies in Africa similarly reported that musculoskeletal issues, particularly back pain, are common complaints among caregivers of children with CP, which are attributed to the demands of caregiving and the lack of assistive devices [38,73–76]. The previous and current study findings underscore the critical need to ensure access to assistive devices, day-care programs, and respite care services to help caregivers find essential relief from the demands of caregiving.

The findings of the present study raise concerns regarding Ethiopia's commitment to the Convention on the Rights of Persons with Disabilities (CRPD), to which it is a signatory. According to Article 20 of the CRPD, Ethiopia is required to ensure personal mobility for individuals with disabilities, including access to assistive devices and technologies. Additionally, Article 9 emphasized the need to eliminate barriers to accessibility in buildings, transportation, and other facilities [77]. Moreover, Ethiopia's commitment to inclusive education is affirmed in its ratification of CRPD. In contrast, in the current study, among the 13 children with CP, only one child was enrolled in school. Children with CP who do not attend school require constant care, making it difficult for their caregivers to manage this duty alongside other responsibilities, such as caring for other family members and managing day-to-day life. This reflects the systemic issue rooted in structural, economic, and attitudinal barriers to inclusive education [51,78], where only 11% of children with special needs are enrolled in primary education and 2.8% in secondary education in Ethiopia (ESDP, 2021). Another research evidence also shows that there is limited acceptance and retention of children with developmental disabilities in Ethiopia's mainstream schools as compared to other disabilities due to their requirement of additional support [52,79]. Therefore, there is an urgent need for the full implementation of CRPD to enhance access to assistive technology, improve the accessibility of physical environments, and promote inclusive education, ultimately reducing the physical strain on caregivers.

Caregivers in our study experienced a range of ongoing emotions from shock and regret to fear and worry about the future, reflecting their children's total dependence, lack of reliable support systems, such as mental health support. This finding is consistent with a recent study conducted in Gondar, which reported that caregivers of children with CP often felt shock, disbelief, fear, and confusion, as well as fear about their child's future [80]. In our study, widespread societal misconceptions surrounding CP led to caregivers' social isolation and contributed to their mental health challenges, including anger, anxiety, forgetfulness, and difficulty concentrating. The finding aligns with previous research, which underscores how caregiving for children with CP can exacerbate mental health challenges, particularly due to future uncertainties and the physical demands of caregiving [28,74]. A study in Nigeria also found that poor marital relationships, lack of social interaction, and feelings of loneliness were linked to depression among caregivers of children with CP [81]. Similarly, caregivers of children with CP in Ghana faced stigmatization, which resulted in social isolation, guilt, and even suicidal thoughts [35]. Our study findings imply the need for ongoing mental health support for caregivers, and creating public awareness campaigns to combat the stigma associated with CP could enhance the well-being of caregivers.

Despite the enormous negative impact of caregiving on caregivers' well-being, positive caregiving experiences were reported, including personal growth and resilience. Caregivers who reported caregiving gains received financial support, training, and job opportunities from CBR and were actively engaged in support groups. These caregivers also used natural support, religion and spirituality, as a source of strength and hope during the challenging time. A previous study in Ethiopia showed spirituality had a vital role in the lives of parents of children with disabilities, especially in the face of discrimination and inadequate support [82]. Another study in Turkey also found that mothers of children with developmental disabilities often turn to spirituality as a coping strategy, helping them manage emotional distress and find meaning in their new lives [83]. A study in Iran also demonstrated that the positive effects of spirituality to reduce mother's caregiving strain [84]. Regarding the benefits of support groups, a study in Ghana indicated that caregivers who participated in support groups showed improved well-being through the realization of 'I'm not alone', improved understanding of the child's condition, a feeling of hope and positive attitudinal changes towards their child and themselves [85]. Overall, the findings indicate that a balanced integration of formal and natural support systems could enhance caregivers' well-being. Future interventions need to prioritize improving caregivers' access to CBR services while recognizing natural support networks, including religious and spiritual institutions. Establishing a collaborative framework between health and rehabilitation professionals and religious leaders could provide sustainable psychological and emotional support for caregivers.

## Support needs of caregivers

Financial support, adequate healthcare service, psychological support, and access to child education were the unmet support needs of caregivers. Even though caregivers want to start a small business and improve their living conditions, financial constraints are a major challenge. In the current study, most caregivers lived in impoverished conditions, with some reporting abandonment by their spouses related to their child's disability. As a result, the priority support need was financial, aimed at economic independence and self-sufficiency. In line with our findings, a similar study in Ethiopia indicated that caregivers of children with developmental disabilities struggled to balance earning income and finding time and resources to support their children. Especially single mothers didn't have the opportunity to work unless they carried their child to work, which placed extra demand [47]. Previous research also identified financial constraints as a major challenge for caregivers of children with disabilities in LMICs due to the high costs of care and limited resources [14,30,75,86]. Another evidence also revealed that caregiving for a child with disability places a substantial financial strain on families, contributing to increased poverty due to direct costs (e.g., medical care) and indirect costs (e.g., lost income from caregivers reducing working hours or leaving jobs) [83,87]. Evidence from South Africa suggests that financial assistance for primary caregivers of children with disabilities helps alleviate immediate financial strain for families [88]. Therefore, providing financial support, such as regular caregiver benefits and livelihood assistance, could help reduce caregivers' economic vulnerability. Moreover, policies need to prioritize establishing and strengthening early childhood development centers and expanding CBR services to enable caregivers' workforce participation and sustainably alleviate their financial burdens.

Our study found that the civil war in the region exacerbated caregivers' economic difficulties. Other researchers reported similar findings highlighting how living in the areas of protracted political conflict increases the inequalities and vulnerabilities of children with disabilities and their caregivers and poses challenges that result in economic loss and diminished wellbeing [89]. A study from Israel showed that war negatively impacted not only the economic status but also psychosocial health of caregivers of children with developmental disabilities. The authors found that during wartime, there were disruptions of educational and rehabilitation services, leading to missed work, unexpected treatment costs, and increased caregiving demands, particularly impacting parents with low economic status [89,90]. Similarly, families and children with disabilities affected by armed conflict in Syria were faced significant challenges, such as losing their homes, livelihoods, and essential resources. These families struggle to meet basic needs, including food, healthcare, housing, assistive devices, medication, therapies, and transportation to access necessary services [91]. Overall, the findings of our and other studies indicate the need to safeguard essential disability-related services during periods of conflict and to provide targeted financial support to economically vulnerable caregivers.

Caregivers had unmet healthcare service needs, including a lack of comprehensive information about CP, long waiting times, unprofessional attitudes from healthcare providers, and limited access to treatments or therapies. Caregivers stated that some healthcare professionals advised them to discontinue their child's follow-up, believing that the child would not develop further. This finding might indicate that the healthcare professionals' attitude is shaped by the medical model of disability, focusing primarily on the child's impairment rather than addressing the holistic and ongoing needs of both the child and caregiver. In addition, caregivers in the current study reported receiving unclear and insufficient information regarding CP, leading to confusion and wasted resources. This finding is consistent with previous studies, which found that caregivers of children with CP often felt inadequately informed and unsupported by healthcare professionals [16,92]. Moreover, negative attitudes from healthcare professionals exacerbated caregivers' feelings of helplessness and intensified their anxieties [18]. Another study conducted in Ethiopia indicated that healthcare providers acknowledged the lack of disability-inclusive clinical care in their professional training [93]. Given that healthcare professionals are often the first point of contact for caregivers, there is a need to strengthen disability-inclusive care by providing targeted training for health providers and integrating disability-related competencies into health professional education programs. Regarding services, although multidisciplinary rehabilitation, including physical rehabilitation, orthopedic surgery, occupational therapy, and psychosocial support is required for CP management [54], caregivers in the present study reported that

their children received only physical therapy. As a result, due to the lack of progress in their child's condition, caregivers reported attending alternative healing options, such as "Holy Water", to get a cure. This reflects caregivers' misconceptions that CP is a curable condition through spiritual practices. Consistent with our study findings, research from Ethiopia has shown that family caregivers of people with chronic health conditions often engage in religious rituals and services to seek miraculous healing and emotional comfort [94]. Therefore, healthcare and rehabilitation professionals need to equip caregivers with clear and accessible health information about the cause and prognosis of CP through educational programs. In a similar context, educational workshops by health care professionals for caregivers of children with CP have improved caregivers' knowledge and health outcomes for the children and their caregivers in Zimbabwe [95,96]. In addition, research has shown that multidisciplinary rehabilitation intervention improved functional motor skills and quality of life, as well as social, emotional, and cognitive outcomes for children with disabilities, and also enhanced satisfaction and helped family caregivers feel more supported and empowered in their care [95,96]. Therefore, it is necessary to ensure access to multidisciplinary rehabilitation services for children with CP. Beyond the health care professionals, training religious leaders and equipping them with accurate information about CP could help address misconceptions and guide caregivers toward appropriate services, as faith leaders are important and trusted members of the community [33,97]. The World Health Organization (WHO) also recognizes the crucial role of faith leaders in supporting the spread of reliable health information and promoting positive health behaviours, specifically in LMICs [98,99].

Overall, although all caregivers in this study were enrolled in the CBR program, significant unmet needs, such as access to assistive devices, psychological support needs, and access to education for their children with CP, were reported. These substantial gaps in services raise critical questions about how the CBR program meets caregivers' needs and what challenges CBR faces in supporting caregivers of children with CP in the Ethiopian context.

## Limitations and future research directions

This study has some limitations. Despite the efforts to include fathers' perspectives, all participants were women. Globally, CP is more prevalent among boys than girls [100–103]; however, the majority of caregivers in our sample were mothers of girls with CP. Additionally, due to security concerns in the region, caregivers were recruited through the CBR program in only one city (Gondar), where the program could facilitate the recruitment process. As a result, the perspectives of caregivers who lack access to CBR services or who live in remote rural areas were not included. Future research should consider a gender-balanced sample and incorporate insights from fathers, siblings, grandparents, and caregivers from rural areas and diverse regions in Ethiopia to gain a comprehensive understanding of family caregiving for children with CP. Moreover, given the sensitive nature of the topic, which involves emotional and intimate conversations, future researchers might consider obtaining verbal consent to enhance the openness and comfort of participants.

## Conclusion

The caregiving experiences of caregivers of children with CP in Ethiopia are characterized by significant physical, psychosocial, and financial challenges, but also resilience. Although caregivers derive support from natural sources, such as religion/spirituality, and family, formal support systems have been found to be insufficient in meeting the needs of caregivers. There is a pressing need for interventions that address caregivers' financial vulnerabilities, enhance access to healthcare and assistive devices, and provide targeted psychosocial support. Moreover, strengthening CBR services to meet caregivers' multifaceted needs and designing interventions that build on caregivers' inherent strengths and resources are essential.

The findings offer insights into caregiving challenges shared across many low- and middle-income countries. Common challenges such as limited formal support, reliance on natural support systems, unmet needs, and the negative impact on caregivers' well-being reflect broader global experiences. The influence of cultural and spiritual beliefs underscored the

importance of contextually sensitive interventions. These insights contribute to a global understanding of caregiving and could inform the development of contextual interventions to enhance caregivers' well-being.

## Supporting information

**S1 File. Interview guide in English.**
(DOCX)

**S1 Checklist. Inclusivity in global research.**
(DOCX)

## Acknowledgments

We would like to extend our gratitude to the study participants and Mr. Zelalem Demeke, the Ethiopian representative of Enablement Foundation (formerly Cerebral Palsy Africa) and an occupational therapist at the University of Gondar, for his invaluable support in recruiting participants. We also thank CBR field workers, who played a crucial role in finding the caregivers. Furthermore, we want to thank Haymanot Ezezew, for her contribution to data collection and Adhanom Baraki (AB) for serving as a secondary coder.

## Author contributions

**Conceptualization:** Melkitu Melak, Beata Batorowicz.

**Formal analysis:** Melkitu Melak, Beata Batorowicz.

**Investigation:** Melkitu Melak.

**Methodology:** Melkitu Melak, Solomon Mekonnen, Afolasade Fakolade, Beata Batorowicz.

**Project administration:** Melkitu Melak.

**Supervision:** Beata Batorowicz.

**Validation:** Solomon Mekonnen, Afolasade Fakolade, Beata Batorowicz.

**Writing – original draft:** Melkitu Melak.

**Writing – review & editing:** Melkitu Melak, Solomon Mekonnen, Afolasade Fakolade, Beata Batorowicz.

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
