## [Decision Letter · Decision Letter 0]

5 Jun 2025

PONE-D-25-19589

“It Feels Like My Spine is About to Break”: Experience and Support Needs of Family Caregivers of Children with Cerebral Palsy in Ethiopia

PLOS ONE

Dear Dr. Melak,

Thank you for submitting your manuscript to PLOS ONE. After careful consideration, we feel that it has merit but does not fully meet PLOS ONE’s publication criteria as it currently stands. Therefore, we invite you to submit a revised version of the manuscript that addresses the points raised during the review process.

We look forward to receiving your revised manuscript.

Kind regards,

Michal Soffer

Academic Editor

PLOS ONE

https://journals.plos.org/plosone/s/file?id=ba62/PLOSOne_formatting_sample_title_authors_affiliations.pdf....

3. Please remove all personal information, ensure that the data shared are in accordance with participant consent, and re-upload a fully anonymized data set.

Additional guidance on preparing raw data for publication can be found in our Data Policy (https://journals.plos.org/plosone/s/data-availability#loc-human-research-participant-data-and-other-sensitive-data) and in the following article: http://www.bmj.com/content/340/bmj.c181.long....

Reviewers' comments:

Reviewer's Responses to Questions

**Comments to the Author**

1. Is the manuscript technically sound, and do the data support the conclusions?

Reviewer #1: Yes

Reviewer #2: Yes

2. Has the statistical analysis been performed appropriately and rigorously?

Reviewer #1: N/A

Reviewer #2: Yes

3. Have the authors made all data underlying the findings in their manuscript fully available?

The PLOS Data policy requires authors to make all data underlying the findings described in their manuscript fully available without restriction, with rare exception (please refer to the Data Availability Statement in the manuscript PDF file). The data should be provided as part of the manuscript or its supporting information, or deposited to a public repository. For example, in addition to summary statistics, the data points behind means, medians and variance measures should be available. If there are restrictions on publicly sharing data—e.g. participant privacy or use of data from a third party—those must be specified. requires authors to make all data underlying the findings described in their manuscript fully available without restriction, with rare exception (please refer to the Data Availability Statement in the manuscript PDF file). The data should be provided as part of the manuscript or its supporting information, or deposited to a public repository. For example, in addition to summary statistics, the data points behind means, medians and variance measures should be available. If there are restrictions on publicly sharing data—e.g. participant privacy or use of data from a third party—those must be specified. requires authors to make all data underlying the findings described in their manuscript fully available without restriction, with rare exception (please refer to the Data Availability Statement in the manuscript PDF file). The data should be provided as part of the manuscript or its supporting information, or deposited to a public repository. For example, in addition to summary statistics, the data points behind means, medians and variance measures should be available. If there are restrictions on publicly sharing data—e.g. participant privacy or use of data from a third party—those must be specified. requires authors to make all data underlying the findings described in their manuscript fully available without restriction, with rare exception (please refer to the Data Availability Statement in the manuscript PDF file). The data should be provided as part of the manuscript or its supporting information, or deposited to a public repository. For example, in addition to summary statistics, the data points behind means, medians and variance measures should be available. If there are restrictions on publicly sharing data—e.g. participant privacy or use of data from a third party—those must be specified.

Reviewer #1: Yes

Reviewer #2: Yes

4. Is the manuscript presented in an intelligible fashion and written in standard English?

Reviewer #1: Yes

Reviewer #2: Yes

5. Review Comments to the Author

Reviewer #1: Thank you for the opportunity to review the manuscript addressing the caregiving experiences of mothers of children with cerebral palsy (CP) in Ethiopia. Findings are important and interesting, shedding light on the lived realities of caregivers in a significantly under-researched context.

Below, I offer several comments and suggestions aimed at further strengthening the manuscript:

1. Literature Review and Framing:

- It would be helpful to distinguish more clearly between findings from HICs and LMICs, both in the introduction and the discussion sections.

- The rationale for the importance of conducting the study in Ethiopia could be expanded and would help justify the study more fully.

- Please clarify whether institutionalization or abandonment of children with CP is prevalent in the Ethiopian context. Since this option was mentioned by some mothers during the interviews, further contextualization would help the reader understand whether this reflects a broader social reality or isolated perceptions.

2. Methods:

- The methodology section is well written, but the introduction could briefly preview the sampling rationale and characteristics (e.g., severe CP, broad age range).

- Further details on the recruitment process (e.g., how caregivers were contacted) would enhance transparency.

- Are any of the children dependent on medical equipment (e.g., respiratory devices)?

- The setting description could be streamlined for focus and clarity.

- Please provide more context for the use of the Multidimensional Model of Caregiving Process and Outcome, which informs the interview guide.

- Consider explaining the predominance of girls with CP in the sample, in contrast with general statistics.

3. Findings:

Where possible, quantify or summarize the prevalence of key themes (e.g., “most parents reported…”), which can help convey the relative weight of each theme and guide the reader through the findings.

In the section on acceptance and spirituality, some subthemes could be grouped under coping. Consider avoiding repetition of quotes or phrasing (e.g., regarding emotional growth or stigma).

When referring to “natural support,” acknowledge that the same support sources may also cause distress, as was reflected earlier in the findings.

4. Discussion:

• The discussion would benefit from clearer articulation of how formal and natural support systems can be balanced.

• The finding that most children were not enrolled in any educational framework deserves further attention. Is this reflective of systemic issues or specific to the sample?

• I recommend that the gendered dimension of caregiving be highlighted more strongly. Since almost all caregivers were mothers, this calls for greater attention to gender-based inequities in care work, and the specific burdens mothers face in resource-limited settings.

• Clarifying the practical relevance of the findings and their significance in the context of global caregiving challenges would help international readers better appreciate the contribution of the study.

• please note that functional classifications such as GMFCS and MACS were determined by parents rather than clinicians, which may limit the precision of these measures.

Reviewer #2: Thank you for submitting the manuscript for my review. The manuscript essentially meets the seven publication criteria required by PLOS ONE. I will first present the strengths of the manuscript’s content, followed by a discussion of its weaker aspects.

The main value of the manuscript submitted for review lies in its empirical foundation: sociological qualitative research on the experiences of caregivers of children with cerebral palsy in Ethiopia.

1. Appropriateness of the methodological approach – the use of an exploratory, descriptive qualitative approach is well justified by the authors. The study addresses a topic that remains underexplored in the Ethiopian context.

2. Well-described data analysis – the use of reflexive thematic analysis (Braun & Clarke), along with a clear description of the coding stages and collaboration with multiple researchers, makes the analysis reliable and trustworthy.

3. Attention to the ethical dimension of the research – the authors ensured the approval of ethics committees, transparency in the recruitment process, and sensitivity to the symbolic competence level of the participants.

4. The efforts made by the authors/researchers to ensure the trustworthiness of the presented data – credibility, dependability, confirmability, and transferability also deserve recognition.

5. The study is grounded in a thoroughly presented socio-cultural context – including the role of religion, traditional gender roles, and spiritual beliefs about disability – as well as in the context of Ethiopia's healthcare system. This allows deeper understanding of the caregivers' situation and enriches the analysis.

6. The structure of the presentation of research findings is logical, clear, and well-organized.

7. It is worth noting that the authors are familiar with the socio-medical concept of “success in illness”, which allowed them to include the caregivers’ positive experiences in their analysis. This balanced approach to the caregivers’ experiences should be recognized and appreciated.

8. The Discussion section is very well written – it demonstrates strong links to the literature, includes critical reflection, and clearly outlines practical implications proposed by the authors.

I will now present the weaker aspects of the manuscript and the research described within it.

Materials and Methods

1. There is no justification provided for the number of study participants (N = 13). The concern is not about the relatively small sample size itself – qualitative methodology does not require large samples – but rather about whether data saturation was achieved. What determined the decision to stop including additional caregivers in the study? According to qualitative methodological standards, this decision should be based on theoretical saturation of the collected material – that is, the point at which no new categories emerge from additional data. Was this the case in the present study? It remains unclear whether and how saturation was reached. The mention of "information power" is vague and does not clarify this point.

2. My next concern relates to the lack of anonymity in the study. The participants may not have perceived their participation as anonymous or confidential, given that they were required to sign written consent forms containing identifying information. This procedure may have reduced the participants’ openness during the interview and weakened the level of trust toward the researcher. Although the consent forms were later anonymized and the participants were assigned numerical identifiers, the act of signing one’s name is not a neutral action – especially when the interview involves emotional, intimate, and difficult topics. For this type of research, verbal consent – recorded together with the interview – would have been more appropriate.

3. The research tool attached (Appendix 1) includes only broadly formulated questions. There is no indication of any follow-up or probing questions the researcher used to explore specific topics in depth.

Findings

4. I have reservations about the way the research findings are presented. Here we have an abundance of quotes and very modest, shallow interpretations by the authors (especially in Topic 1). Although the use of the participants’ direct statements strengthens credibility, deepens the analysis, and superbly illustrates the emotions and context, the authors’ commentary is too sparse and lacks analytical depth. Description clearly dominates here, without deeper conceptualizations. The interpretative potential of the collected qualitative material should be utilized more thoroughly.

5. The presented research findings also lack individual differences, descriptions of various cases beyond the main themes, which contradicts the standards for reporting qualitative data (e.g., COREQ checklist, 2007, item 32). The caregivers’ experiences may vary depending on the sociodemographic characteristics, family situation, or attitude toward religion. While reading the findings, I wondered about the answers to the following questions: Does the level of caregivers’ education influence their understanding of the diagnosis? Does the family situation (e.g. divorce) affect the adaptation process to the child’s disability? Does having more children influence the process of coming/the ability to come to terms with the child’s illness? Was faith in God a facilitating factor in coping with the child’s disability for every caregiver? Was this true for every caregiver who declared themselves as Orthodox Christians? In the context of care for a child with CP, does the birth order of the affected child matter? Did the ongoing civil war in the study region (Amhara) impact the participants’ experiences (for example, through worsening of their economic situation and causing additional stress)? I am convinced that data on secondary themes/experiences of the caregivers could make the analyses more diverse and even more interesting. Highlighting individual differences could successfully broaden and deepen the authors’ interpretations, as qualitative research is not about just citing respondent statements, which in the text are numerous, lengthy, and therefore overrepresented.

6. The analysis of the empirical material could be deepened by incorporating field notes, which the authors only briefly mention. From the text, we know that the researcher made them after the interviews, but we do not know whether their content was consistent with the interviews. I am curious if they included the researcher’s observations regarding the caregivers’ health functioning (visible spinal problems, possible difficulties with walking, or overall physical fitness). How does the lack of assistive equipment (e.g. standing frames, hoists) affect the caregivers’ physical health? It is impossible to conduct an interview without also performing participant observation.

I assess the text as valuable. The presented research findings identifying the specific needs of caregivers of children with CP are very interesting and undoubtedly have practical value for designing support for this group of parents in Ethiopia. In my opinion, the authors should be given the opportunity to publish the text after implementing the recommended corrections and addressing the above remarks.

Recommendations for Development of the Research Project:

1. Due to the highly diverse social structure of Ethiopia, in my opinion, the research should be continued and extended to rural populations or other regions of the country. The recruitment was limited to participants from a city covered by a single CBR program (Gondar), which weakens the transferability of the results.

2. Although mothers are the main caregivers, their exclusive participation may limit insight into family care processes. The study could benefit from including fathers, grandparents, or siblings of children with CP.

6. PLOS authors have the option to publish the peer review history of their article (what does this mean?). If published, this will include your full peer review and any attached files.). If published, this will include your full peer review and any attached files.). If published, this will include your full peer review and any attached files.). If published, this will include your full peer review and any attached files.

**Do you want your identity to be public for this peer review?** For information about this choice, including consent withdrawal, please see our  For information about this choice, including consent withdrawal, please see our  For information about this choice, including consent withdrawal, please see our  For information about this choice, including consent withdrawal, please see our Privacy Policy....

Reviewer #1: No

Reviewer #2: No

---

## [Author Response · Author response to Decision Letter 1]

28 Aug 2025

Date: 2025/08/22

To: Editor, PLOS ONE

Response to reviewers’ comments

We would like to thank you and the reviewers for the valuable feedback on ways to improve and strengthen our manuscript, titled: “It Feels Like My Spine is About to Break”: Experience and Support Needs of Family Caregivers of Children with Cerebral Palsy in Ethiopia. Below, we respond to each point raised by the reviewers, indicating any changes and their location in the manuscript with track changes. We have also attached a clean manuscript without track changes. Thank you for your consideration. We look forward to your response.

Reviewer One

1. Literature review and Framing:

It would be helpful to distinguish more clearly between findings from HICs and LMICs, both in the introduction and the discussion sections.

Thank you for your valuable feedback. We have revised the introduction as you suggested and presented the findings for HICs and LMICs separately. (Page 4-5/line 77-99)

The rationale for the importance of conducting the study in Ethiopia could be expanded and would help justify the study more fully.

Comment accepted and we revised the rationale to as follows: “A recent scoping review on the health outcomes and support needs of caregivers of children with CP in SSA highlighted a lack of research from Ethiopia, despite a growing body of evidence in the region (Melak et al., 2025). While previous studies from other countries could provide some insight, the unique socio-cultural and economic factors, such as the spiritual and collectivist nature of Ethiopian society (Karbo, 2013), multidimensional poverty (UNDP., 2022a), and stigma (Zuurmond et al, 2022) might influence the caregiving experience differently. Moreover, the limited availability of disability services and formal caregiving support systems in Ethiopia could impact caregiving for children with CP. To address this knowledge gap, the current study explored the caregiving experiences and support needs of family caregivers for children with CP in Ethiopia. Such a study is needed to provide a foundational evidence base to inform future research, as well as give insight into the design of relevant policies and support interventions tailored to the specific needs and realities of caregivers of children with CP in Ethiopia.” (Page 5-6/line 105-118)

Please clarify whether institutionalization or abandonment of children with CP is prevalent in the Ethiopian context. Since this option was mentioned by some mothers during the interviews, further contextualization would help the reader understand whether this reflects a broader social reality or isolated perceptions.

We appreciate your concern. We want to clarify that while some of our participants noted that they were initially considering the option of placing their child for adoption upon learning about their child's condition for the first time, they explained that this their initial reaction, largely due to a lack of information and support. However, none of the participants ultimately pursued this option; instead, they came to view their child's condition as a gift from God over time. Furthermore, we found no evidence of abandonment among children with CP within the Ethiopian context. (Page 17/line 309-312)

2. Method:

The methodology section is well written, but the introduction could briefly preview the sampling rationale and characteristics (e.g., severe CP, broad age range).

Comment accepted, and we edited accordingly: “Evidence showed that the most significant predictors of caregiver burden were the duration of caregiving and the level of dependency of the care recipient (Lindt et al.). Caregivers of children with CP were found to spend, on average, 15 hours per day in Turkey (Park & Nam, 2019) and 21.3 hours per day in Ethiopia (Kassa et al., 2024) on caregiving activities. Furthermore, mothers of children with CP with higher scores on the GMFCS tend to spend more time on caregiving than mothers of children with lower GMFCS scores (Ahmadi Kahjoogh et al., 2016). They also experience higher levels of stress due to increased caregiving demands, health concerns, and potential developmental delays. Similarly, as the child with CP gets older and heavier, the physical demand placed on caregivers also increases (Dambi et al., 2015).” (Page 3/ line 63-64 & page 4/ line 67-71)

Further details on the recruitment process (e.g., how caregivers were contacted) would enhance transparency.

Based on your suggestion, we added the following: “Caregivers who were recruited for the study were contacted by CBR field workers to inquire about their willingness to participate in the research.” (Page 8/ line 183-185)

Are any of the children dependent on medical equipment (e.g., respiratory devices)?

No, none of the caregivers had a child with CP who uses medical equipment such as a respiratory device.

The setting description could be streamlined for focus and clarity.

We appreciate the feedback. We kept it as it was, as we think that offering a detailed description of the study setting is important to help readers better contextualize the findings and assess their potential transferability to similar settings.

Please provide more context for the use of the Multidimensional Model of Caregiving Process and Outcome, which informs the interview guide.

We accepted the comment, and we have edited as follows to provide more context: “The interview guide was developed based on the Multidimensional Model of Caregiving Process and Outcome, developed by Raina et al. (2004), which provides a comprehensive understanding of the complex interplay between various factors that influence the caregiving experience. This model highlights how background/contextual factors (e.g., socioeconomic status), caregiver strain, intrapsychic factors and coping factors interact to affect caregiver outcomes such as physical and psychological health. The model informed the development of the interview guide, helping us identify key areas to explore, including caregivers’ day-to-day experiences, challenges, coping strategies, available support systems, and the broader contextual influences on caregiving” (Page 13/ line 206-213)

Consider explaining the predominance of girls with CP in the sample, in contrast with general statistics.

Thank you for your suggestion. We added this under the limitations of the study: “Globally, CP is more prevalent among boys than girls (Ahmed et al., 2023; Chounti et al., 2013; Yang et al., 2021; Yeargin-Allsopp et al., 2008); however, the majority of caregivers in our sample were mothers of girls with CP. Future research should consider a gender-balanced sample and incorporate insights from fathers, siblings, grandparents, and caregivers from rural areas and diverse regions in Ethiopia to gain a comprehensive understanding of family caregiving for children with CP”. (Page 41/ line 872-873 & Page 42/ line 877-879)

3. Findings: Where possible, quantify or summarize the prevalence of key themes (e.g., “most parents reported…”), which can help convey the relative weight of each theme and guide the reader through the findings.

Thank you for your feedback. While we acknowledge the value of knowing the prevalence, we want to clarify that for this study, we followed the specific approach of reflexive thematic analysis (RTA) (Braun & Clarke, 2006, 2019, 2020). To make our work congruent with this approach, we generated rich, nuanced accounts of meaning rather than measured or quantified theme frequency. RTA views themes as patterns of shared meaning, rather than categories to be counted (Braun & Clarke, 2006, 2019, 2020).

In the section on acceptance and spirituality, some subthemes could be grouped under coping. Consider avoiding repetition of quotes or phrasing (e.g., regarding emotional growth or stigma).

We greatly appreciate your feedback and noticing some repetition. We have now integrated/moved the acceptance subtheme into the natural support subtheme, specifically integrating with the findings of spirituality as coping, as caregivers cited religion and spirituality as their reasons for accepting their child's condition. (Page 31/ line 617-635)

When referring to “natural support,” acknowledge that the same support sources may also cause distress, as was reflected earlier in the findings.

We appreciate the feedback. We edited accordingly: From: “Caregivers received physical, informational, and emotional support from their natural support systems, such as family, friends, and neighbours.”

To: “Caregivers received physical, informational, and emotional support from their natural support systems, such as family, friends, and neighbours. However, the same support sources were reported as a source of distress in the earlier findings.” (Page 29/ line 573-575)

4. Discussion:

The discussion would benefit from clearer articulation of how formal and natural support systems can be balanced.

We appreciate the suggestion. We have revised the discussion section:

From: “The findings of the current study imply that a balanced formal and natural support could enhance the positive experience of caregivers of children with CP.”

To: “The findings of the current study suggest that a balanced integration of formal and natural support could enhance caregivers' positive experiences. Therefore, it is crucial to improve the accessibility and availability of CBR support services. Recognizing the role of natural support systems and the roles of religion and spirituality in addressing the psychological and emotional needs of caregivers could have a sustainable impact on the well-being of caregivers. Therefore, fostering a system of collaboration between health professionals and religious leaders is vital.” (Page 39/ line 814-820)

The finding that most children were not enrolled in any educational framework deserves further attention. Is this reflective of systemic issues or specific to the sample?

Comment accepted and revision has been made accordingly. We have added to the discussion section as follows: “Moreover, Ethiopia’s commitment to inclusive education is affirmed in its ratification of CRPD. In contrast to this, in the current study, among the 13 children with CP, only one child was enrolled in school. This reflects the systemic issue rooted in structural, economic, and attitudinal barriers to inclusive education (Franck & Joshi, 2017; Tekola et al., 2020), where only 11% of children with special needs are enrolled in primary education and 2.8% in secondary education in Ethiopia (ESDP, 2021). Research evidence also shows that there is limited acceptance and retention of children with developmental disabilities in Ethiopia’s mainstream schools as compared to other disabilities due to their requirement of additional support (Okyere et al.; Tilahun et al., 2016). This evidence implies that the underdeveloped social services and systemic challenges impose an additional burden on caregivers. Therefore, the evidence highlights the urgent need for the full implementation of CRPD to enhance access to assistive technology, improve the accessibility of physical environments, and promote inclusive education, ultimately reducing the physical strain on caregivers.” (Page 38/ line 779-791)

I recommend that the gendered dimension of caregiving be highlighted more strongly. Since almost all caregivers were mothers, this calls for greater attention to gender-based inequities in care work, and the specific burdens mothers face in resource-limited settings.

Recommendation accepted, and we have added to the discussion section as follows: “Globally, deep-rooted gender norms result in family caregiving still being widely perceived as women’s work (Stall et al., 2023). As a result, women currently perform up to 81% of unpaid caregiver roles worldwide (Bhan et al., 2020). In the current study, all caregivers were women who assumed the responsibility of meeting their children's daily needs, managing household chores, and sometimes serving as the primary breadwinners. This finding highlights that caregiving responsibilities disproportionately fall on women, emphasizing that future interventions should consider the gendered inequities involved in caring for children with CP. Contrary to our finding, a recent study conducted in Saudi Arabia found that caregiving for children with CP was shared between fathers and mothers, indicating a shift from the traditional view of women as the primary caregivers (Alqahtani et al., 2025). ” (Page 37/ line 755-763)

Clarifying the practical relevance of the findings and their significance in the context of global caregiving challenges would help international readers better appreciate the contribution of the study.

Thank you for the feedback. We added the following to the conclusion:

“The findings offer insights into caregiving challenges shared across many low- and middle-income countries. Common challenges such as limited formal support, reliance on natural support systems, unmet needs and negative impact on the well-being of caregivers reflect broader global experiences. The influence of cultural and spiritual beliefs underscored the importance of contextually sensitive interventions. These insights contribute to a global understanding of caregiving and can inform the development of context-responsive policies and support practices.” (Page 42/ line 893-898)

Please note that functional classifications such as GMFCS and MACS were determined by parents rather than clinicians, which may limit the precision of these measures.

Although GMFCS, MACS, and CFCS classifications were based on caregivers’ perceptions, we minimized potential misclassification by providing clear guidance on the meaning of each level.

We have modified from: GMFCS, MACS, and CFCS tools were used to classify the functional ability of children with CP as perceived by caregivers.

To this: GMFCS, MACS, and CFCS tools were used to classify the functional abilities of children with CP based on caregivers’ perceptions, with guidance provided on the meaning of each level. (Page 9/ line 202-203)

Reviewer 2

Materials and Methods

1.There is no justification provided for the number of study participants (N = 13). The concern is not about the relatively small sample size itself – qualitative methodology does not require large samples – but rather about whether data saturation was achieved. What determined the decision to stop including additional caregivers in the study? According to qualitative methodological standards, this decision should be based on theoretical saturation of the collected material – that is, the point at which no new categories emerge from additional data. Was this the case in the present study? It remains unclear whether and how saturation was reached. The mention of "information power" is vague and does not clarify this point.

We greatly appreciate the concern. To clarify, consistent with our approach (exploratory qualitative descriptive), we were guided by the model of information power (the more relevant information a sample holds, the fewer participants are needed) (Malterud et al., 2015) to determine the adequacy of sample size. This model suggests that the size of a sample with sufficient information power depends on the study's aim, sample specificity, use of established theory, quality of dialogue, and analysis strategy. Our study had a focused aim, the sample was specific, the interview guide was informed by the Multidimensional Model of Caregiving Process and Outcome, and interviews were conducted in-depth by an experienced interviewer in the participants’ native language, Amharic. These characteristics align with the principles of information power, making a sample size of 13 sufficient to address the research question.

2. My next concern relates to the lack of anonymity in the study. The participants may not have perceived their participation as anonymous or confidential, given that they were required to sign written consent forms containing identifying information. This procedure may have reduced the participants’ openness during the interview and weakened the level of trust toward the researcher. Although the consent forms were later anonymized and the participants were assigned numerical identifie

---

## [Decision Letter · Decision Letter 1]

6 Oct 2025

Dear Dr. Melak,

Thank you for submitting your manuscript to PLOS ONE. After careful consideration, we feel that it has merit but does not fully meet PLOS ONE’s publication criteria as it currently stands. Therefore, we invite you to submit a revised version of the manuscript that addresses the points raised during the review process.

The manuscript has been evaluated by two reviewers, and their comments are available below. The reviewers have raised a number of concerns that need attention. Could you please revise the manuscript to carefully address the concerns raised?

We look forward to receiving your revised manuscript.

Kind regards,

Johanna Pruller, Ph.D.

Senior Editor

PLOS ONE

Journal Requirements:

Reviewers' comments:

Reviewer's Responses to Questions

**Comments to the Author**

Reviewer #1: All comments have been addressed

Reviewer #2: All comments have been addressed

2. Is the manuscript technically sound, and do the data support the conclusions?

Reviewer #1: Yes

Reviewer #2: Yes

3. Has the statistical analysis been performed appropriately and rigorously?

Reviewer #1: N/A

Reviewer #2: N/A

4. Have the authors made all data underlying the findings in their manuscript fully available?

Reviewer #1: Yes

Reviewer #2: Yes

5. Is the manuscript presented in an intelligible fashion and written in standard English?

Reviewer #1: Yes

Reviewer #2: Yes

Reviewer #1: I would like to thank you again for the opportunity to review this revised manuscript. The authors have seriously and thoughtfully addressed the comments raised in the previous review round, and the revisions have considerably strengthened the paper. At the same time, I believe there remain several areas that would benefit from further clarification and elaboration:

1. Authers stated that "While previous studies from other countries could provide some insight, the unique socio-cultural and economic factors, such as the spiritual and collectivist nature of Ethiopian society (Karbo, 2013), multidimensional poverty (UNDP., 2022a), and stigma (Zuurmond et al, 2022) might influence the caregiving experience differently"- the term “stigma” remains too general and vague. It would be helpful to clarify what kind of stigma is referred to, and to whom.

2. Disability services and formal caregiving support – The mention of the limited availability of services and support systems in Ethiopia is important, but somewhat underdeveloped. I encourage the authors to either provide supporting references or expand on concrete details regarding the types of services lacking and how this impacts caregiving for children with CP.

3. Impact of conflict and instability – The authors added valuable examples of how the ongoing conflict and instability in the region affect economic challenges. However, I wondered whether there are more specific examples relevant to caregiving itself (e.g., access to rehabilitation, transportation difficulties, or disruptions to medical or educational services).

4. Moreover, if the authors address the critical nuance of caregiving in a conflict zone (which I think adds complexity and relevance), I recommend expanding the background and/or discussion to include theoretical or empirical literature on the impact of protracted political conflict or war on quality of life and caregiving for children with disabilities, including CP. This would add depth and situate the findings within a broader scholarly framework.

Reviewer #2: The author responded to all comments in a thorough and systematic manner, citing relevant literature and revising the manuscript to address missing elements (e.g., probing questions, explanation of limitations, the role of field notes, and the impact of regional conflict). It is clear that all issues raised by the reviewer were taken seriously and are reflected in the revised version of the manuscript.

Strengths of the response:

Clear reference to each comment,

Implementation of changes in the manuscript according to the reviewer’s suggestions,

Deep reflection on ethical procedures,

Expanded methodology section and discussion of study limitations.

Weaknesses:

No explicit statement that data saturation was achieved; instead, the author relies solely on the concept of “information power,” which does not fully meet the reviewer’s expectations,

Limited response to concerns about “shallow interpretation”—rather than deepening the analysis, the author primarily defended the chosen methodological approach (exploratory-descriptive).

Recommendation:

The responses are convincing and should satisfy the editors; however, it may be worth considering:

Adding a brief note stating that after a certain number of interviews no new categories emerged (which would suggest that saturation was indeed achieved),

Adding 2–3 additional sentences in the discussion to further elaborate on interpretation, for example by linking the findings to the broader literature to demonstrate that the analytical potential of the data was fully utilized within the chosen approach.

.

Reviewer #1: No

Reviewer #2: No

---

## [Author Response · Author response to Decision Letter 2]

2 Dec 2025

Reviewer #1

1. Authers stated that "While previous studies from other countries could provide some insight, the unique socio-cultural and economic factors, such as the spiritual and collectivist nature of Ethiopian society (Karbo, 2013), multidimensional poverty (UNDP., 2022a), and stigma (Zuurmond et al, 2022) might influence the caregiving experience differently"- the term “stigma” remains too general and vague. It would be helpful to clarify what kind of stigma is referred to, and to whom.

Response: Thank you for the constructive feedback, we have made revisions accordingly.(In clean manuscriptPage 5/ line 96)

2. Disability services and formal caregiving support – The mention of the limited availability of services and support systems in Ethiopia is important, but somewhat underdeveloped. I encourage the authors to either provide supporting references or expand on concrete details regarding the types of services lacking and how this impacts caregiving for children with CP.

Response: Thank you for the valuable feedback! We have added details about the types of services with supporting references.(Page 5/from line 98-102)

3. Impact of conflict and instability – The authors added valuable examples of how the ongoing conflict and instability in the region affect economic challenges. However, I wondered whether there are more specific examples relevant to caregiving itself (e.g., access to rehabilitation, transportation difficulties, or disruptions to medical or educational services).

Response: Thank you for your relevant feedback. However, aside from economic challenges, we did not find additional conflict-related caregiving experiences in our data.

4. Moreover, if the authors address the critical nuance of caregiving in a conflict zone (which I think adds complexity and relevance), I recommend expanding the background and/or discussion to include theoretical or empirical literature on the impact of protracted political conflict or war on quality of life and caregiving for children with disabilities, including CP. This would add depth and situate the findings within a broader scholarly framework.

Response: Thank you for the valuable insight. In response to your suggestion, we added a paragraph that discusses our findings related to conflict in relation to the existing literature(Page 39/from line 819-831)

Reviwer #2

1.No explicit statement that data saturation was achieved; instead, the author relies solely on the concept of “information power,” which does not fully meet the reviewer’s expectations,

Response: We appreciate your valuable feedback regarding saturation. However, as we outlined below, based on established methodological evidence and in line with similar exploratory qualitative descriptive studies that cite information power as a guiding principle, we justified our sample size using information power rather than data saturation.

• First, Braun and Clarke emphasize that saturation is a concept originating from Grounded Theory, where data collection continues until no new information emerges. This idea, introduced by Glaser and Strauss, is tied to the constant comparison method in which each new observation is compared with previous analytical insights. Because saturation is embedded in the logic of Grounded Theory, it is not conceptually aligned with other analytic approaches(Braun & Clarke, 2019).

• Accordingly, Braun and Clarke (2019) argue that saturation is unsuitable for reflexive thematic analysis and recommend information power as a more appropriate framework for determining sample adequacy. Information power shifts the focus from “Have I stopped hearing anything new?” to “Does my sample hold sufficient, relevant, and meaningful information to address the study aim?” It emphasizes richness, depth, and relevance of the data rather than redundancy of responses.

• This aligns well with exploratory qualitative descriptive approach, where the goal is not to capture every possible dimension of a phenomenon (as in theoretical saturation), but rather to provide varied and sufficiently rich accounts. Thus, data sufficiency, rather than saturation, is the more appropriate criterion.

• Information power suggests that the adequacy of a sample depends on the richness, depth, and relevance of the data in relation to the study aim rather than on reaching a point where no new themes appear. Instead of asking, “When do I stop hearing anything new?”, information power asks, “Does my sample hold enough meaningful information to answer my research question?”(Malterud et al., 2016)

• Given that the aim of exploratory qualitative descriptive research is not to capture every possible dimension of a phenomenon (as in theoretical saturation), but rather to provide varied and sufficiently rich accounts (Hunter David et al., 2019), our focus was on data sufficiency rather than saturation.

• Our study was therefore guided by the concept of information power(Malterud et al., 2016).Recent exploratory qualitative descriptive studies in health and rehabilitation sciences have adopted this approach to justify sample adequacy(Engeda et al.; Gaurav et al., 2023; Vuuren et al., 2023).

• Following this methodological guidance, we assessed our sample in relation to the study aim, the specificity of samples, the quality of the dialogue, and the richness of the narratives obtained. Data collection was concluded after 13 interviews, as the accounts were sufficient to answer the research questions.(Page 8/ line 174-178 and 185-186)

2.Limited response to concerns about “shallow interpretation”—rather than deepening the analysis, the author primarily defended the chosen methodological approach (exploratory-descriptive).

Response: We appreciate this concern. We have revised the discussion section to provide a deeper interpretation of the findings by more explicitly situating them within the broader global evidence(Page 34-42)

3.Adding a brief note stating that after a certain number of interviews no new categories emerged (which would suggest that saturation was indeed achieved),

Response: We have already described our justification above.(Page 8/ line 174-178 and 185-186)

4. Adding 2–3 additional sentences in the discussion to further elaborate on interpretation, for example by linking the findings to the broader literature to demonstrate that the analytical potential of the data was fully utilized within the chosen approach.

Response: Thank you for the insightful feedback. We have revised the discussion and more clearly connected our findings to the broader literature.

(Page 35/ line 722-727,

744-746,

764-768,

786-792,

795-797,

Page 39/from line 819-831

Page 40/ line 844-847,

851-853,

856-858,

)

---

## [Decision Letter · Decision Letter 2]

5 Feb 2026

Dear Dr. Melak,

Thank you for submitting your manuscript to PLOS ONE. After careful consideration, we feel that it has merit but does not fully meet PLOS ONE’s publication criteria as it currently stands. Therefore, we invite you to submit a revised version of the manuscript that addresses the points raised during the review process.

We look forward to receiving your revised manuscript.

Kind regards,

Helen Howard

Staff Editor

PLOS One

Journal Requirements:

Reviewers' comments:

Reviewer's Responses to Questions

**Comments to the Author**

Reviewer #1: All comments have been addressed

Reviewer #2: All comments have been addressed

2. Is the manuscript technically sound, and do the data support the conclusions?

Reviewer #1: Yes

Reviewer #2: Yes

3. Has the statistical analysis been performed appropriately and rigorously?

Reviewer #1: N/A

Reviewer #2: N/A

4. Have the authors made all data underlying the findings in their manuscript fully available?

Reviewer #1: Yes

Reviewer #2: Yes

5. Is the manuscript presented in an intelligible fashion and written in standard English?

Reviewer #1: Yes

Reviewer #2: Yes

Reviewer #1: The revised manuscript demonstrates improvement compared to the previous version. The discussion is more comprehensive and better grounded in the literature, and the authors provide clearer, more reasonable explanations of the study's methodological aspects. However, several minor issues still require further clarification, as outlined below.

1. In the statement, “In this context, caregiving could impose a double burden on women who already have demanding day-to-day roles and might also exclude them from participation in all areas of life, such as education, employment, and decision-making, which in turn affect their ability to provide optimal care for their child with CP,”

the authors are encouraged to clarify what is meant by “decision-making.” Specifically, it would be helpful to distinguish whether this refers to household decision-making, healthcare-related decisions, or broader social and community-level participation. In addition, the authors should more explicitly explain the mechanisms through which exclusion from decision-making may influence the quality of care provided to children with CP.

2. The issue of providing accurate psychoeducational information and addressing misconceptions is highly important; however, this section would benefit from clearer articulation. Specifically, the authors are encouraged to clarify how the proposed strategies (e.g., multidisciplinary rehabilitation, professional guidance, and training religious leaders) contribute to improving caregivers’ understanding and satisfaction. In addition, it would strengthen the discussion to support these recommendations with references to empirical studies demonstrating that such approaches have been effective in similar contexts.

3. A proofreading of the Discussion section is required, as it contains several language and phrasing issues. (e.g., the phrase “Contrary to study” is one example and should be revised to “Contrary to the present study” or “In contrast to the findings of this study”); punctuation should be added where necessary etc.

Reviewer #2: The authors’ responses are satisfactory and adequately address all concerns raised. The revisions substantially strengthen the manuscript, both in terms of methodological justification and the depth of interpretation. Therefore, I recommend the manuscript for publication.

.

Reviewer #1: No

Reviewer #2: No

---

## [Author Response · Author response to Decision Letter 3]

1 Mar 2026

1. In the statement, “In this context, caregiving could impose a double burden on women who already have demanding day-to-day roles and might also exclude them from participation in all areas of life, such as education, employment, and decision-making, which in turn affect their ability to provide optimal care for their child with CP,”

the authors are encouraged to clarify what is meant by “decision-making.” Specifically, it would be helpful to distinguish whether this refers to household decision-making, healthcare-related decisions, or broader social and community-level participation. In addition, the authors should more explicitly explain the mechanisms through which exclusion from decision-making may influence the quality of care provided to children with CP.

Response: Thank you for the constructive feedback. We have clearly updated the statement as recommended. From this:

“In this context, caregiving could impose a double burden on women who already have demanding day-to-day responsibilities and might also exclude them from participation in all areas of life, such as education, employment, and decision-making, which in turn affects their ability to provide optimal care for their child with CP.”

Into this:

“In this context, caregiving could impose a double burden on women who already have demanding day-to-day responsibilities and might also exclude them from participation in education, paid employment, and political and social engagement (46, 71). Such exclusion could reduce their access to resources, autonomy in household and healthcare decision-making, and impact their well-being, ultimately affecting their ability to provide quality care for their child with CP.”

2. The issue of providing accurate psychoeducational information and addressing misconceptions is highly important; however, this section would benefit from clearer articulation. Specifically, the authors are encouraged to clarify how the proposed strategies (e.g., multidisciplinary rehabilitation, professional guidance, and training religious leaders) contribute to improving caregivers’ understanding and satisfaction. In addition, it would strengthen the discussion to support these recommendations with references to empirical studies demonstrating that such approaches have been effective in similar contexts.

Response: We have clearly articulated the statement and also provided supporting evidence.

From this: “Therefore, it is necessary to ensure access to comprehensive multidisciplinary rehabilitation therapy to maximize the functional outcomes for children with CP and enhance caregivers' satisfaction with services. In addition, health care professionals need to provide caregivers with accurate, clear information about CP. Training religious leaders could also facilitate the dissemination of information to caregivers (94).”

Into this: “……This reflects caregivers’ misconceptions that CP is a curable condition through spiritual practices. Consistent with our study findings, research from Ethiopia has shown that family caregivers of people with chronic health conditions often engage in religious rituals and services to seek miraculous healing and emotional comfort (95). Therefore, healthcare and rehabilitation professionals need to equip caregivers with clear and accessible health information about the cause and prognosis of CP through educational training programs. In a similar context, educational workshops by health care professionals for caregivers of children with CP have improved caregivers' knowledge and health outcomes for the children and their caregivers in Zimbabwe (96, 97). In addition, research has shown that multidisciplinary rehabilitation intervention improved functional motor skills and quality of life, as well as social, emotional, and cognitive outcomes for children with disabilities, and also enhanced satisfaction and helped family caregivers feel more supported and empowered in their care (96, 97). Therefore, it is necessary to ensure access to multidisciplinary rehabilitation services for children with CP. Beyond the health care professionals, training religious leaders and equipping them with accurate information about CP could help address misconceptions and guide caregivers toward appropriate services, as faith leaders are important and trusted members of the community (33, 98). The World Health Organization (WHO) also recognizes the crucial role of faith leaders in supporting the spread of reliable health information and promoting positive health behaviours, specifically in LMICs (99, 100).”

3. A proofreading of the Discussion section is required, as it contains several language and phrasing issues. (e.g., the phrase “Contrary to study” is one example and should be revised to “Contrary to the present study” or “In contrast to the findings of this study”); punctuation should be added where necessary etc.

Response: We have thoroughly read and corrected the phrasal and grammatical issues, as well as the punctuation, throughout the discussion section.

---

## [Editor Report · Decision Letter 3]

16 Mar 2026

“It Feels Like My Spine is About to Break”: Experience and Support Needs of Family Caregivers of Children with Cerebral Palsy in Ethiopia

PONE-D-25-19589R3

Dear Dr. Melak,

We’re pleased to inform you that your manuscript has been judged scientifically suitable for publication and will be formally accepted for publication once it meets all outstanding technical requirements.

Kind regards,

Taiwo Opeyemi Aremu, MD, MPH, PhD

Academic Editor

PLOS One
---

## [Editor Report · Acceptance letter]

PONE-D-25-19589R3

PLOS One

Dear Dr. Melak,

I'm pleased to inform you that your manuscript has been deemed suitable for publication in PLOS One. Congratulations! Your manuscript is now being handed over to our production team.

Kind regards,

on behalf of

Dr. Taiwo Opeyemi Aremu

Academic Editor

PLOS One